# Tryptase β regulation of joint lubrication and inflammation via proteoglycan-4 in osteoarthritis

Nabangshu Das[1,2,3,4], Luiz G. N. de Almeida[2,3,4,5], Afshin Derakhshani[2,3,4,5], Daniel Young[2,3,4,5], Kobra Mehdinejadiani[2,3,4,5], Paul Salo [3], Alexander Rezansoff[1,3], Gregory D. Jay[6], Christian P. Sommerhoff[7], Tannin A. Schmidt[3,8], Roman Krawetz [3,9] ✉ & Antoine Dufour [1,2,3,4,5,10] ✉

PRG4 is an extracellular matrix protein that maintains homeostasis through its boundary lubricating and anti-inflammatory properties. Altered expression and function of PRG4 have been associated with joint inflammatory diseases, including osteoarthritis. Here we show that mast cell tryptase β cleaves PRG4 in a dose- and time-dependent manner, which was confirmed by silver stain gel electrophoresis and mass spectrometry. Tryptase-treated PRG4 results in a reduction of lubrication. Compared to full-length, cleaved PRG4 further activates NF-κB expression in cells overexpressing TLR2, −4, and −5. In the destabilization of the medial meniscus model of osteoarthritis in rat, tryptase β and PRG4 colocalize at the site of injury in knee cartilage and is associated with disease severity. When human primary synovial fibroblasts from male osteoarthritis patients or male healthy subjects treated with tryptase β and/or PRG4 are subjected to a quantitative shotgun proteomics and proteome changes are characterized, it further supports the role of NF-κB activation. Here we show that tryptase β as a modulator of joint lubrication in osteoarthritis via the cleavage of PRG4.

Osteoarthritis (OA) is a complex and heterogeneous disease involving diverse inflammatory cells, loss of boundary lubrication and the mechanical breakdown of articular cartilage, however, the connection between these factors remains poorly characterized[1–3]. Proteoglycan 4 (PRG4) is an extracellular glycoprotein synthesized and secreted by cells that line the joint cavity (i.e., synoviocytes and superficial zone chondrocytes), where it contributes to homeostasis by providing boundary lubrication[4–6]. In addition to its role in lubrication, PRG4 can also regulate inflammation through interaction with different cell surface receptors, including CD44[7], Toll-like receptors (TLRs)[8,9], and L-selectin[10]. PRG4 contains different domains of functional and structural importance; a central mucin-like domain flanked by a cysteine-rich N-terminal somatomedin-like domain, and a C-terminal hemopexin-like domain[11]. For boundary lubrication, the N-terminus, C-terminus, or both facilitate its attachment to the articulating surfaces by interacting with other macromolecules[12], resulting in

[1]Faculty of Kinesiology, University of Calgary, Calgary, AB, Canada. [2]Hotchkiss Brain Institute, Cumming School of Medicine, University of Calgary, Calgary, AB, Canada. [3]McCaig Institute for Bone and Joint Health, Cumming School of Medicine, University of Calgary, Calgary, AB, Canada. [4]Snyder Institute for Chronic Diseases, Cumming School of Medicine, University of Calgary, Calgary, AB, Canada. [5]Department of Biochemistry and Molecular Biology, Cumming School of Medicine, University of Calgary, Calgary, AB, Canada. [6]Department of Emergency Medicine, Warren Alpert Medical School & School of Engineering, Brown University, Providence, RI, USA. [7]Institute of Medical Education and Institute of Laboratory Medicine, University Hospital, LMU Munich, Munich, Germany. [8]Biomedical Engineering Department, University of Connecticut Health Center, Farmington, CT, USA. [9]Cell Biology and Anatomy, Cumming School of Medicine, University of Calgary, Calgary, AB, Canada. [10]Physiology and Pharmacology, Cumming School of Medicine, University of Calgary, Calgary, AB, Canada. ✉e-mail: rkrawetz@ucalgary.ca; antoine.dufour@ucalgary.ca

protein-protein interactions with PRG4 that is folded with the mucin domain[13]. The central mucin domain, which is extensively glycosylated with GalNAc-Gal cores with a negatively charged sialic acid caps, repels through steric repulsion when articulating surfaces compress during movement[6,13]. In various tissues and fluids, PRG4 is found as monomers, disulfide-bonded dimers, and multimers with a molecular weight ranging from 223 to 1493 kDa[14]. While the presence of sialylated and sulfated structures on the mucin domain of PRG4 is assumed to contain epitopes that interact with L-selectin on the surface of peripheral and synovial polymorphonuclear neutrophils[10], the interaction of the C-terminal domain with CD44 is reported to alter pro-inflammatory NF-κB expression[7]. Currently, little is known if different conformations of PRG4 (monomers vs. multimers) have overlapping or distinct biological functions and if the biological functions of PRG4 are impacted by post-translational modifications such as proteolysis[15].

Altered expression and functions of PRG4 have been associated with acute and chronic diseases, including OA[16], rheumatoid arthritis (RA)[16], Sjögren's syndrome[15,17,18], atherosclerosis[19], and pericarditis[20,21]. The *Prg4−/−* mice demonstrated cartilage degeneration which resulted in a defect in skeletal morphology[22]. We and others have demonstrated that PRG4 injections into injured joints of mice[22], rats[8], and pigs[23] resulted in a significant delay in OA progression. However, PRG4 treatment does not completely stop the progression of OA. There are additional mechanisms that must impact how PRG4 acts within an inflammatory OA environment. A better understanding of PRG4's functions and its post-translational modifications could help better explain these observations.

Tryptase β is a serine protease[24] and is the most abundant secretory granule protein in human mast cells (MCs)[25,26]. Tryptase β has been predominantly studied for its key pathogenic functions in asthma[26,27], and there are ongoing clinical trials (NCT04092582) studying the effect of tryptase β inhibition with an anti-tryptase antibody for the treatment of severe asthma[28]. MCs play pivotal roles in the regulation of host response and tissue homeostasis through the release of potent signaling molecules such as histamine and tryptase β, and have been only recently implicated in OA[29,30]. There is growing evidence that MCs play a key role in knee OA and when mast cells were depleted or their activities were blocked, the progression of OA was delayed[29]. Over 8500 OA knee samples were analyzed and the use of $H_1$-antihistamines was associated with reduced prevalence of knee OA[31]. These preclinical results have gained traction because of the association of antihistamine usage with decreased OA severity in patient populations[31]. These findings are particularly interesting since they resemble the effects observed in pre-clinical models of OA treated with PRG4[8,22,23]. Therefore, we hypothesized that proteolytic regulation of PRG4 by tryptase β could play a pathogenic role in OA. Here, we show that tryptase β degrades recombinant human PRG4 resulting in a loss of lubrication. Secreted PRG4 interacts with TLRs on the cell surface to activate NF-κB. Moreover, we show evidence that tryptase β can cleave endogenous PRG4 in healthy human synovial fluid. Our study provides a rationale for developing a tryptase β inhibitor in combination with intra-articular injections of PRG4 as a therapeutic agent for OA.

## Results

### Proteolytic processing of PRG4 by tryptase β

We hypothesized that PRG4 is susceptible to proteolytic processing and we explored if tryptase β was able to cleave PRG4 as increased tryptase β activity and mast cells are abundant in OA-affected joints[29,32]. Enzyme kinetic analyses of recombinant human proteins using an SDS-PAGE gel followed by silver staining (Fig. 1a) revealed that human PRG4 was efficiently cleaved by tryptase β within 5 min of incubation at 37 °C generating fragments of ~50 kDa (Fig. 1b). The cleaved PRG4 form was stable for at least 240 min. (Fig. 1b). PRG4 was also cleaved by tryptase in a dose-dependent manner when incubated for 1 hour at 1:10, 1:100, and 1:1000 tryptase β:PRG4 molar ratios

(Fig. 1c). PRG4 cleavage by tryptase β was prevented by the serine protease inhibitor, AEBSF, at a dose of 1, 10, and 100 μM (Fig. 1d). As the mucin domain of PRG4 is extensively glycosylated and is known to provide its boundary lubrication property, we wanted to explore if a change in glycosylation would impact tryptase β processing of PRG4. PRG4 and tryptase β were incubated for 1 h or 18 h after being subjected to two deglycosylation enzymes: PNGase F, to remove N-linked glycans, and an *Enteroccocus faecalis* O-glycosidase, to remove O-linked glycans. Deglycosylation of the processed PRG4 fragments resulted in complete degradation of the ~50 kDa PRG4 fragments and generated additional lower molecular weight fragments (Supplementary Fig. 1a). Therefore, glycosylation of PRG4 decreases its susceptibility to be processed by tryptase β.

Next, to determine the N-terminal sequences of the potential multiple protein fragments that we detected, we used a mass spectrometry approach called Amino-Terminal Oriented Mass Spectrometry of Substrates (ATOMS)[33,34] (Fig. 1e). A mixture of PRG4 was dimethylated with light formaldehyde ($CH_2O$, +28 Da), while a mixture of PRG4 and tryptase β was dimethylated with heavy formaldehyde ($CD_2O$, +34 Da). Both mixtures were combined before being subjected to LC-MS/MS followed by MaxQuant[35,36] analysis (Fig. 1e). After a 1 min incubation, six cleavage sites were identified: $^{146}K↓K^{147}$, $^{147}K↓V^{148}$, $^{201}K↓S^{202}$, $^{303}K↓T^{304}$, $^{984}K↓V^{985}$, and $^{1306}R↓R^{1307}$, (Fig. 1f). After a 5 min incubation, eight cleavage sites were identified: $^{146}K↓K^{147}$, $^{147}K↓V^{148}$, $^{251}K↓I^{252}$, $^{303}K↓T^{304}$, $^{1224}R↓D^{1225}$, $^{1284}R↓R^{1285}$, $^{1330}K↓G^{1331}$, and $^{1345}K↓G^{1346}$ (Fig. 1g). Using ATOMS to identify cleavage sites of PRG4 by tryptase β at 1 h and 24 h incubation, we found 28 and 31 cut sites, respectively (Supplementary Fig. 1b, c), suggesting that tryptase β can degrade PRG4 in vitro. Thus, we identified PRG4 as a substrate of tryptase β.

### Reduced PRG4 lubricating ability after tryptase β cleavage

PRG4 is a glycoprotein with important lubricating functions[5,18,37]. We next examined if processed PRG4 would result in a loss of lubrication. Using a tribology test with a glass-polydimethylsiloxane (PDMS) polymer interface[38], we performed a velocity sweep analysis at different lubricating regimes (Fig. 2a). Our tribological results demonstrated that the addition of tryptase β to PRG4 causes loss of boundary lubrication (low velocity) and mixed lubrication (medium velocity) but not hydrodynamic lubrication (high velocity) (Fig. 2b, Supplementary Fig. 2). This loss could be partially rescued when AEBSF was added to the mixture to inhibit tryptase activity. The loss of PRG4 lubrication was observed after 1 h, 6 h, and 24 h addition of tryptase (Fig. 2c). Thus, processing of PRG4 by tryptase β resulted in a loss of lubricating ability.

### Tryptase β cleavage of PRG4 associates with articular cartilage degeneration

To study the impact of PRG4 processing by tryptase β, we used an OA-induced destabilization of the medial meniscus (DMM) rat model[39] that resembles clinical meniscal injury in humans (Fig. 3a). Utilizing three different antibodies (tryptase β (red), antibody to PRG4 mucin domain (blue), and antibody to PRG4 C-terminal (green)) and DAPI (white) (Fig. 3b), we stained rat knee joints over a period of 1 thru 4 weeks post-DMM. Interestingly, no signal for the PRG4 C-terminal antibody was detected in the sham-operated control cartilage, but the PRG4 mucin domain antibody showed intense staining of the superficial layer of the cartilage as expected[40] (Fig. 3c). One-week post-DMM, loss of mucin domain PRG4 with the concurrent appearance of C-terminal PRG4 and Tryptase was observed on the surface of articular cartilage. We also detected a disrupted/abnormal expression of mucin and C-terminal PRG4 in the cartilage along with colocalization of tryptase β in the areas where PRG4 was detected (Fig. 3d, Supplementary Fig. 3a, b). Two weeks post-DMM, diminished PRG4 and tryptase β staining was detected (Fig. 3e, Supplementary Fig. 3c). The mucin domain antibody to PRG4 (blue) was undetectable at three weeks post-DMM (Fig. 3f, Supplementary Fig. 3d) but by four weeks

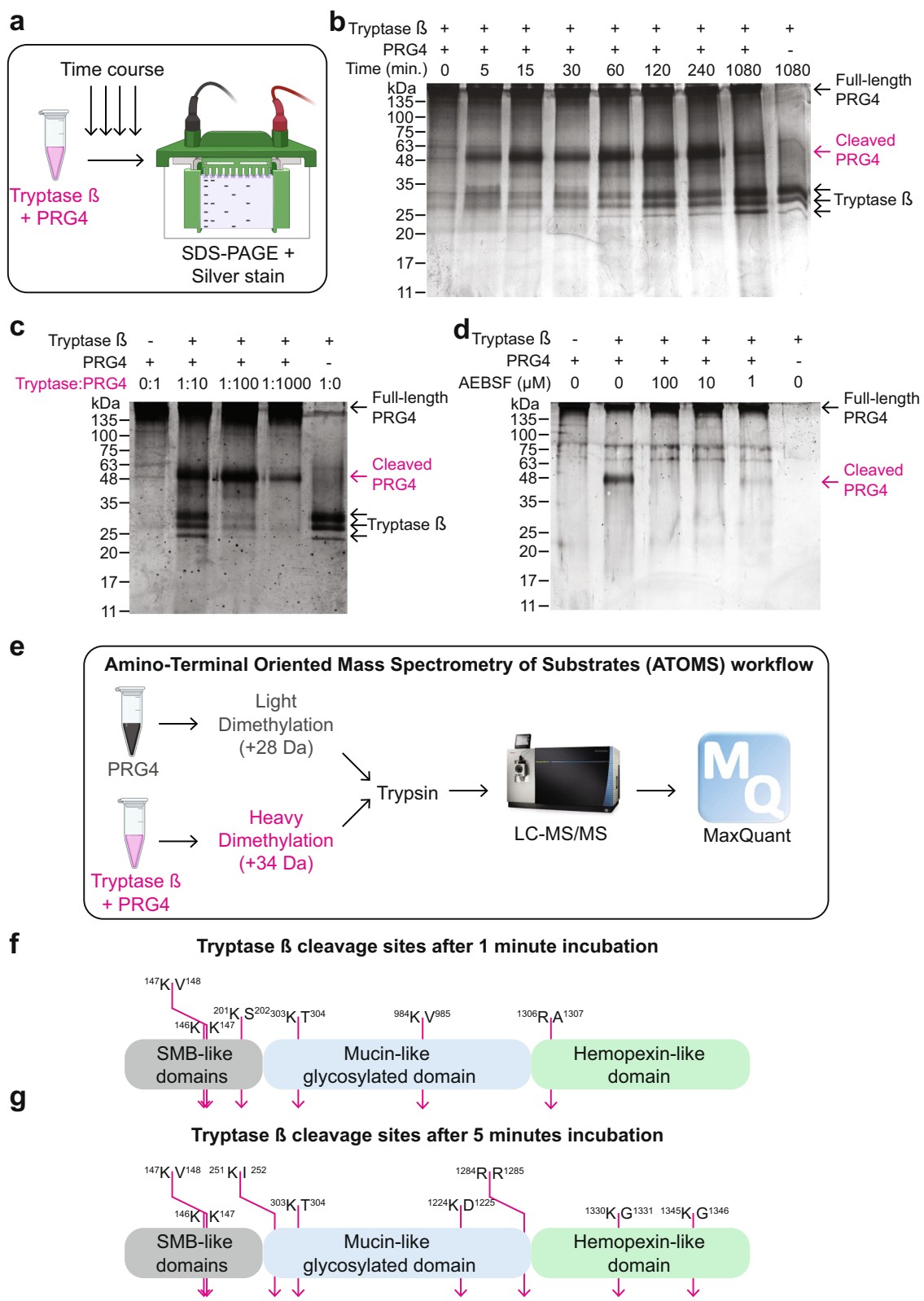

post-DMM, reactivity to the mucin domain antibody was re-observed at the cartilage surface (Fig. 3g, Supplementary Fig. 3e).

Next, we wanted to see if we could prevent this loss of PRG4 in the DMM model by injecting recombinant PRG4 at the time of DMM surgery (Fig. 3h, Supplementary Fig. 4a). Four weeks post-DMM, increased PRG4 staining was detected on the cartilage surface of rats treated with recombinant PRG4 as compared to control saline injections (Fig. 3i, j and Supplementary Fig. 4a–c). Therefore, injections of PRG4 could

partially prevent this loss of PRG4 levels. Also, the localization of tryptase β and PRG4 corresponded to regions of cartilage injury and areas where PRG4 was lost.

## Tryptase β cleavage of PRG4 enhances NF-κB activity via TLR receptors

We previously demonstrated that PRG4 binds directly to TLR2,−4, and −5 and activates NF-κB[8]. Next, we investigated if tryptase β processing

**Fig. 1 | Investigation of tryptase β-processed PRG4. a** Schematic of the experimental workflow (created with BioRender). **b** Silver-stained 10% SDS-PAGE gel of in vitro cleavage of human recombinant PRG4 (initial mass of PRG4 1 µg) by human recombinant tryptase β (1:100 enzyme (tryptase)/substrate (PRG4)) incubated at 5, 15, 30, 60, 120, 240 and 1080 min at 37 °C. Magenta arrow indicates cleaved PRG4. This is a representative image of an experiment that was repeated four times with similar results. **c** Silver-stained 10% SDS-PAGE gel of in vitro cleavage of PRG4 by tryptase at 1:10, 1:100 and 1:1000 of enzyme (tryptase)/substrate (PRG4) ratio incubated for 60 min at 37 °C. Magenta arrow indicates cleaved PRG4. This is a representative image of an experiment that was repeated four times with similar results. **d** Silver-stained 10% SDS-PAGE gel of in vitro cleavage of PRG4 (initial mass of PRG4 1 µg) by tryptase β at 1:100 of (tryptase)/substrate (PRG4) ratio incubated

for 60 min at 37 °C with AEBSF, a serine protease inhibitor at 1, 10 and 100 µM. Magenta arrow indicates cleaved PRG4. This is a representative image of an experiment that was repeated four times with similar results. **e** Schematic of the ATOMS workflow (created with BioRender). PRG4 was incubated for 5 min. before being subjected to light formaldehyde labeling ($CH_2O$, +28 Da). Tryptase β and PRG4 were incubated for 5 min. before being subjected with deuterated/heavy formaldehyde ($CD_2O$, +34 Da). Both samples were mixed, and trypsin was added before being analyzed on an LC-MS/MS. RAW files data were analyzed by MaxQuant. **f** Eight different cleavage sites were identified between the heavy dimethylated sample (Tryptase β + PRG4) in comparison to the light dimethylated sample (Tryptase β + PRG4 + AEBSF) after 1 min incubation or **g** 5 min incubation.

of PRG4 changes its ability to activate NF-κB. We used a HEK blue media reporter cells system that either lack TLRs (TLR-null) or have overexpression of TLR2, −4, or −5 (Fig. 4a). TNFα was used as a positive control. In the TLR-null cells, no activation of NF-κB was detected, except for the positive control TNFα ($p < 0.001$), suggesting that PRG4 requires a TLR for NF-κB activation in this cell type (Fig. 4b). In TLR2$^+$ cells, addition of PRG4 + tryptase β resulted in a significant increase of NF-κB when compared to PRG4 ($p < 0.001$) or TNFα ($p = 0.03$) (Fig. 4c). Addition of PRG4 + tryptase β in the presence of AEBSF significantly decreased NF-κB activation ($p < 0.001$) (Fig. 4c). In TLR4$^+$cells, no significant difference was detected between PRG4 + tryptase β and TNFα, but PRG4 + tryptase β significantly activated NF-κB as compared to PRG4 ($p < 0.001$) and PRG4 + tryptase β + AEBSF ($p < 0.001$) (Fig. 4d). In TLR5$^+$cells, PRG4 + tryptase β was able to activate NF-κB above the levels observed in the TNFα ($p = 0.03$), PRG4 ($p < 0.001$) or PRG4 + tryptase β + AEBSF ($p < 0.001$) groups (Fig. 4e). Therefore, tryptase β processing of PRG4 resulted in a significant increase of NF-κB activation as compared to full length PRG4 and this activation was achieved either via TLR2, −4, or −5.

### PRG4 is expressed in synovial lining fibroblasts

OA presents a dynamic cellular landscape throughout the disease course; therefore, we wanted to know which specific cell types within the joints are selectively expressing *PRG4*, *TLR2*, −4, and −5. We mined the literature for rodent arthritis models and used a dataset with the associated number of GSE184609 for single-cell RNA sequencing (scRNA-seq) analysis[41]. Population distribution of scRNA-seq data of hind paw joint cells isolated at the indicated time points of glucose-6-phosphate isomerase (GPI)-induced arthritis were determined. After excluding low-quality cells, our new analysis included 24,496 cells from 5 mice. Uniform manifold approximation and projection (UMAP) based clustering separated cells into 10 individual clusters (Fig. 5a and Supplementary Fig. 5a–c). These clusters predominantly belonged to three classes, namely fibroblasts, monocytes/neutrophils, and macrophages (Fig. 5a). We next analyzed the percentages of cell types in each group. We found that the abundance of various cell types differs widely between naive, Day 6, 14, and 25. We detected a reduced number of cells between fibroblasts and synovial lining fibroblasts between naïve and the GPI-arthritis-induced groups (Day 14 vs. naive) (Fig. 5b–d). Interestingly, *Prg4* was explicitly expressed specifically in synovial lining fibroblasts as seen previously[42] (Fig. 5c). Subsequently, we looked at the expression pattern of *Prg4*, *Thy1(CD90)*, *Tlr2*, *Tlr4*, and *Tlr5*, and identified *Tlr4* as a predominant gene in the synovial lining fibroblasts, whereas *Tlr2* and *Tlr5* appear to be expressed by other cell types (Fig. 5c). Additionally, *Pgr4's* expression is downregulated over time from the naive group to Day 6, 14, and 25 (Fig. 5e). As we have shown previously[7], PRG4 is known to interact with other cell surface receptors such as CD44 (Fig. 5f); therefore, we next investigated the crosstalk between synovial lining fibroblasts and other cell types in the hind paw joint cells using Squidpy. The level of interaction between lining fibroblasts and monocytes/neutrophils was the most elevated in the naive group (Fig. 5g). On Day 14, the most significant

communications between synovial lining fibroblasts and macrophages and monocytes/neutrophils were mediated via the CD44/HBEGF pairs (Fig. 5h). Overall, in a mouse model of inflammatory arthritis, our data analysis suggests that PRG4 could signal via TLR4 and/or CD44 to activate NF-κB.

### Comparison of the proteomes of non-OA vs OA primary human synovial cells

To characterize the biological differences occurring in OA, we isolated primary human synovial CD90$^+$/THY1$^+$ cells using cell sorting (Fig. 6a). Our human studies were carried out in adherence to the principles of the Declaration of Helsinki. As these fibroblasts were demonstrated in mice to have minimal to no PRG4 expression[41] (Fig. 5c), we subjected them to the addition of recombinant PRG4, PRG4 + tryptase β, tryptase β, or buffer alone. In parallel, we used a similar CD90$^+$ cell sorting approach in OA patients (Fig. 6a). All five different conditions ($n = 3$ biological replicates, and 2 technical replicates) were subjected to a quantitative shotgun proteomics workflow using tandem mass tags (TMT) 6-plex labeling (Fig. 6a). After LC-MS/MS analysis, database search for the identification of peptides and proteins was performed with MaxQuant[35] at 1% false discovery rate (FDR). Statistical analysis was performed with MSstatsTMT[43]. We identified 2,206 proteins and quantified 2,166 proteins (Supplementary Table 1). We found a loss of PRG4 in OA patients' cells as compared to buffer-treated non-OA cells. As expected, we found elevated levels of PRG4 in the PRG4-treated cells (Fig. 6b), but these levels were reduced when PRG4 was added in combination with tryptase β, which is consistent with our data that tryptase β degrades PRG4 (Fig. 1f, Supplementary Fig. 1c). In a similar fashion to OA patients' cells, multiple proteins levels were reduced when PRG4 was added with tryptase β, including phosphatidylinositol-binding clathrin assembly protein (PICALM), inositol 1,4,5-trisphosphate receptor type 3 (ITPR3), hepatoma-derived growth factor (HDGF), alpha-L-iduronidase (IDUA), and CD44 (Fig. 6b). Interestingly, HDGF was previously reported to exert mitogenic activity on fibroblasts and PRG4 to regulate proliferation of fibroblasts[44,45], therefore, suggesting a loss of this activity in OA or when tryptase β was added with PRG4[46,47]. Interestingly, there was also a loss of ossification components (SLC39A14, CCDC154, THBS3, IGF2, MMP14, RBP4) in OA fibroblasts when compared to buffer-treated non-OA fibroblasts, as analyzed by Metascape[48] (Fig. 6c, Supplementary Fig. 11a, b). Using Metascape, the OA fibroblasts had an enrichment of signaling by interleukins and expressed a significant increase in the interleukin-13 receptor subunit alpha-2 (IL13RA2) and both cell groups shared an enrichment for cell-cell adhesion (Fig. 6c, Supplementary Fig. 11a, b). When non-OA fibroblasts were treated with PRG4 + tryptase, an enrichment of retrograde endocannabinoid signaling (GNB3, ANXA6, PDHA1, NDUFB3, FGF2, GNB1) and mesenchyme development (DPPA4, PEF1, ANXA6) was identified over those treated with PRG4, using Metascape (Fig. 6d–f). Genome-wide pathway-based association analysis identified retrograde endocannabinoid signaling pathway associated with inflammatory arthritis[49]. Also, enrichment of genes associated with mesenchyme development including DPPA4 was

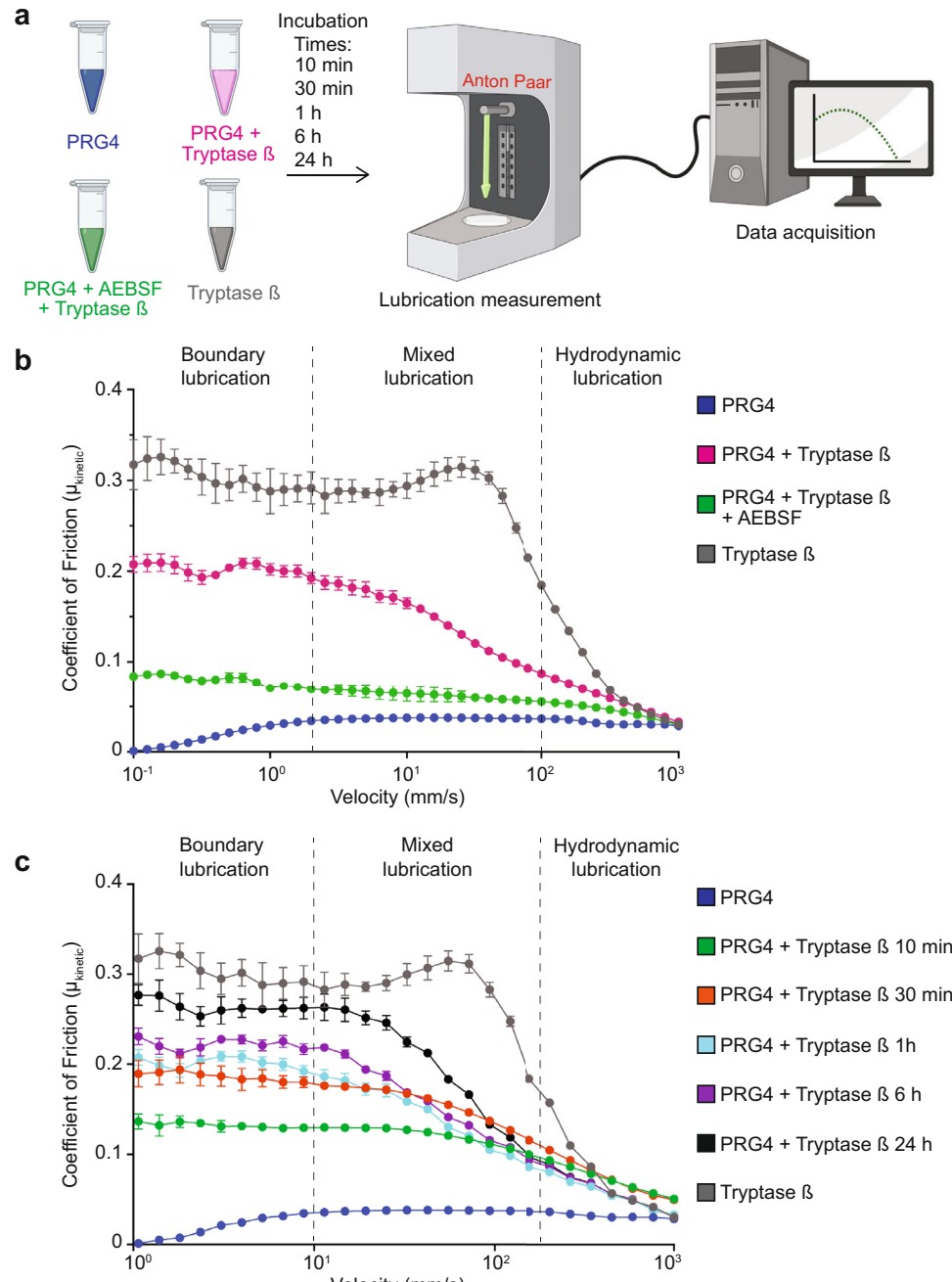

**Fig. 2 | Investigation of the lubrication of PRG4 and tryptase β-processed PRG4.** **a** Schematic of the experimental workflow of the lubrication test (created with BioRender). **b** Lubrication of tryptase β (grey), PRG4 (initial mass of PRG4 1 μg) (blue), PRG4 + tryptase β for 18 h (1:100 enzyme/substrate) (magenta), PRG4 + tryptase β + 1 mM AEBSF incubated for 18 h (1:100 enzyme/substrate) (green). The error bars denote the error of the mean ± standard deviation as obtained from 5 independent experiments. **c** Lubrication of PRG4 (initial mass of PRG4 1 μg) (blue), PRG4 + tryptase β incubated for 10 min (green), 30 min (orange), 1 h (pale blue), 6 h (purple), 24 h (black) and tryptase β (grey). The error bars denote the error of the mean ± standard deviation as obtained from 5 independent experiments.

associated with changes in phenotypes of cellular niche during early OA[50]. Overall, tryptase processing of PRG4 impact the proteomes of primary healthy human fibroblast promoting a phenotype that resemble OA phenotypes.

## Tryptase β cleaves PRG4 in human synovial fluid

The molecular mechanisms implicated in OA initiation and progression remain elusive[51]. Importantly, PRG4 is found in high concentrations in the synovial fluid of joints (~400 μg/ml)[52], which often changes in OA[5,6]. As proteases play key biological roles in synovial fluid and in OA[53–55], we investigate if tryptase β can cleave PRG4 in complex ex vivo

synovial fluid. As the collection of synovium fluid is challenging and limited for healthy patients, collected synovial fluid from cadaveric normal joints (n = 7) within 4 h of death, with no evidence of cartilage pathology on dissection, and incubated them with tryptase or vehicle (Fig. 7a). Proteins from healthy synovial fluid treated with vehicle were labeled with light formaldehyde (+28 Da dimethylation), while synovial fluid incubated with tryptase β for 1 h were labeled with heavy formaldehyde (+34 Da dimethylation) (Fig. 7a). To identify the cleavage sites (neo-N-termini), we subjected the treated synovial fluid to an N-terminomics/TAILS protocol, where the N-termini are enriched using the dendritic polyglycerol aldehyde TAILS polymer[56] (Fig. 7a). The

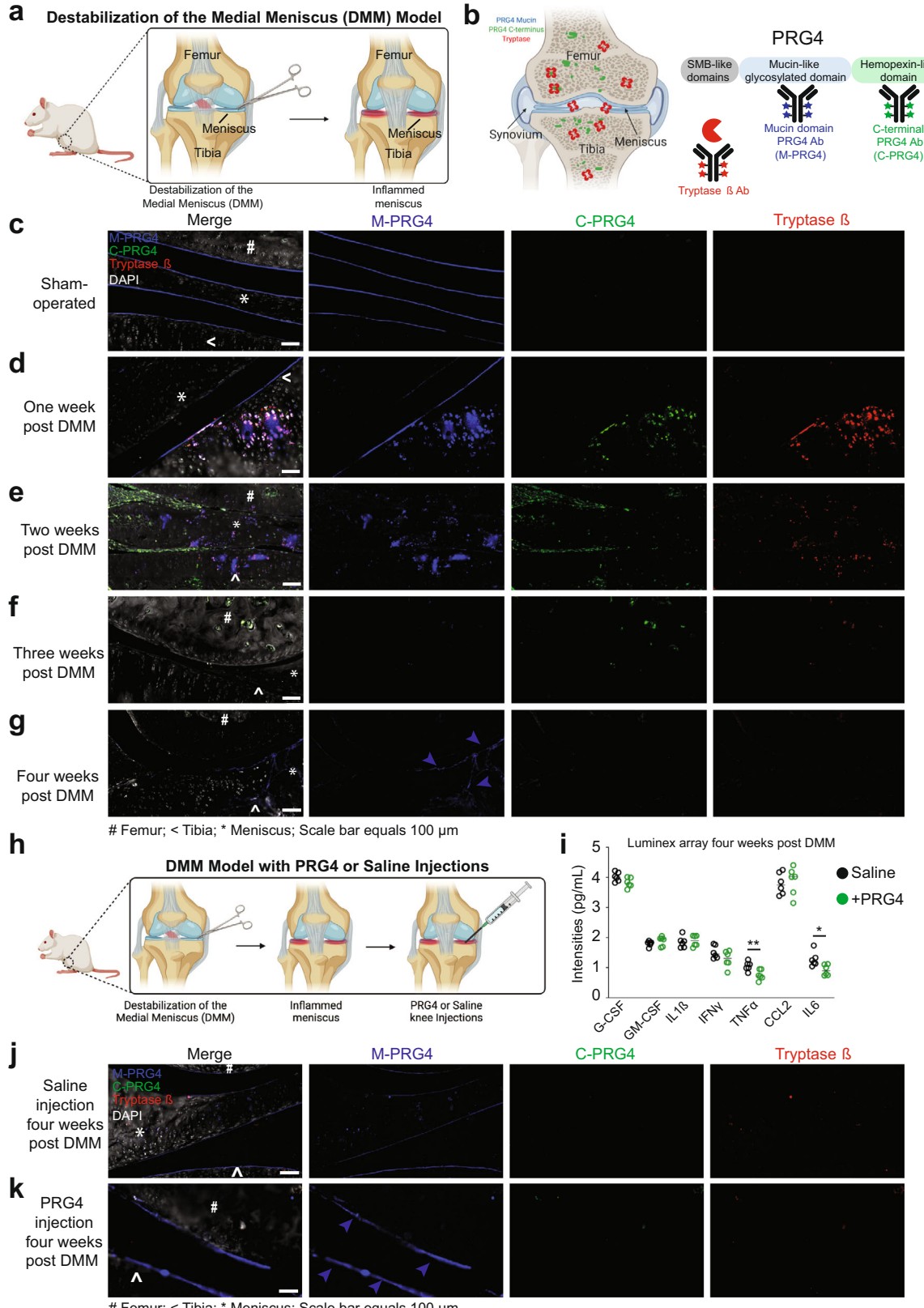

# Femur; < Tibia; * Meniscus; Scale bar equals 100 μm

# Femur; < Tibia; * Meniscus; Scale bar equals 100 μm

global proteomes were compared by a shotgun (pre-enrichment TAILS) proteomics analysis[57] (Fig. 7a). After sample acquisition LC-MS/MS analysis, data were analyzed using MaxQuant[35] at 1% FDR. Shotgun/preTAILS analysis of seven healthy synovial fluid samples incubated with vehicle or tryptase yielded 2524 unique peptides corresponding to 1498 unique proteins, and the TAILS analysis yielded 310 unique

peptides, where 80 were identical to the pre-enrichment TAILS analysis (Fig. 7b, Supplementary Tables 2–6). In the preTAILS data, we identified a significant change of 4.8% of peptides in the tryptase β-treated synovial fluid and 5.8% in the buffer control (Fig. 7c, Supplementary Tables 2, 3). We next analyzed the N-terminal processing in the tryptase-treated samples and identified predominantly internal

**Fig. 3 | Investigation of Tryptase β and PRG4 in the DMM rat model. a** Schematic of the experimental workflow of the destabilization of the medial meniscus (DMM) model (created with BioRender). **b** Schematic of the three antibodies used for immunofluorescence: tryptase β (red), mucin domain of PRG4 (blue), C-terminal domain of PRG4 (green) (created with BioRender). DAPI is show in white. PRG4 and tryptase staining in rat joints in **c** uninjured, **d** one-week post-DMM, **e** two weeks post-DMM, **f** three weeks post DMM, and **g** four weeks post-DMM (n = 15 total, n = 3 per groups). Scale bar = 100 μm. # indicates femur, <indicates tibia and * indicates

the meniscus. **h** Schematic of the experimental workflow of saline and recombinant PRG4 injection (200 μg/kg) in the DMM model. **i** Multiplex analysis (Luminex xMAP technology) of 7 cytokines/chemokines (G-CSF, GM-CSF, IFNγ, IL1β, TNFα CCL2, and IL6) was used to profile rat serum treated with saline or PRG4 (n = 6 per group). PRG4 and tryptase staining in rat joints four weeks post-DMM in **j** saline- or **k** PRG4-injected animals (n = 3 per groups, repeated twice). Scale bar = 100 μm. # indicates femur, <indicates tibia and * indicates the meniscus.

N-termini (87.3%), in addition to other proteoforms, including signal peptide removal (7.7%) and alternative start sites (5.5%) (Fig. 7d, Supplementary Tables 4–6). Next, we generated IceLogos to determine cleavage site preferences between tryptase β-treated and buffer-treated synovial fluids[58]. As expected, we identified a preference for P1 lysine and arginine residues, and a preference for glycine in P1′ and P2′ position in the tryptase β-treated group (Fig. 7e). In the buffer-treated groups, we found a preference for P1 tryptophan and arginine residues (Fig. 7f). In the TAILS data of tryptase β-treated synovial fluid, we identified a cleavage site of PRG4 at position $^{1330}K\downarrow G^{1331}$ (Fig. 7g, Supplementary table 6), which was similarly identified by our ATOMS experiment (Fig. 1g). We additionally identified 14 new potential tryptase substrates including CXCL7/PPBP, ORM1, vimentin, fibrinogen alpha/beta/gamma chains, SAA1, SAA2, APOC3, APOE, fibronectin, gelsolin, sulfhydryl oxidase 1, and procollagen C-endopeptidase enhancer 1 (Fig. 7h, Supplementary table 6). Using Metascape, we generated a protein-protein interaction networks by merging all significantly changed proteins (Supplementary Fig. 8b) and found minimal overlap between the tryptase β and buffer-treated cells (Supplementary Fig. 8a, c). Using STRING-db[59], we identified an enrichment of the activation of C3 and C5, ECM proteoglycans, and toll-like receptor cascade in the buffer-treated synovial fluid (Supplementary Fig. 8d). When analyzing the pre-enrichment TAILS data, we identified an enrichment of cytokine signaling in immune system and adaptive immune response in buffer that is lost in tryptase-treated synovial fluids, which suggests an effect on the N-termini processing as identified the TAILS data (Fig. 7k). Therefore, we identified new potential human tryptase β substrates and validated that PRG4 is an endogenous tryptase β substrate.

## Cleaved PRG4 in human synovial fluid of OA patients

As PRG4 is found in high concentrations in the synovial fluid of joints (~400 μg/ml)[52] and it is often changed in OA[5,6], we wanted to compare the N-terminome of OA patients and healthy subjects and investigate if we could identify cleaved PRG4 in OA. Collected synovial fluid from cadaveric normal joints (n = 3) within 4 h of death, with no evidence of cartilage pathology on dissection, was compared to the synovial fluids of OA patients (Fig. 8a). Proteins from healthy synovial fluid treated with vehicle were labeled with light formaldehyde (+28 Da dimethylation), while synovial fluid incubated with tryptase β for 1 h were labeled with heavy formaldehyde (+34 Da dimethylation) (Fig. 8a). To identify the cleavage sites (neo-N-termini), we subjected the healthy and OA synovial fluids to an N-terminomics/TAILS protocol, where the N-termini are enriched using the dendritic polyglycerol aldehyde TAILS polymer[56] (Fig. 8a). The global proteomes were compared by a shotgun (pre-enrichment TAILS) proteomics analysis[57] (Fig. 8a). After sample acquisition LC-MS/MS analysis, data were analyzed using MaxQuant[35] at 1% FDR. Shotgun/preTAILS analysis of three healthy and three OA synovial fluid samples yielded 2,222 unique peptides corresponding to 920 unique proteins, and the TAILS analysis yielded 452 unique peptides, where 21 were identical to the pre-enrichment TAILS analysis (Fig. 8b, Supplementary Tables 7–11). In the preTAILS data, we identified a significant change of 4.8% of peptides in the OA synovial fluid and 5.6% in the healthy group (Fig. 8c, Supplementary Tables 7, 8). We next analyzed the N-terminal processing in the OA samples and identified predominantly internal N-termini (66.6%), in addition to

other proteoforms, including signal peptide removal (33.3 (Fig. 8d, Supplementary Tables 9–11). In the TAILS data of OA synovial fluid, we identified a cleavage site of PRG4 at position $^{1306}K\downarrow A^{1307}$ (Fig. 8e, Supplementary table 11), which was similarly identified by our ATOMS experiment (Fig. 1f). We additionally identified 20 cleaved proteins in OA including IGF1, IGF2, ECM1, APOB, TIMD4, TIMP2, GIG25, SOD3, SERPINC1, FGA, SAA1, PROC, HRG, C3, APCS, HPX, PRG4, SEPP1, C4A, CFB (Fig. 8f, Supplementary table 11). Using Metascape, we generated a protein-protein interaction networks by merging all significantly changed proteins (Supplementary Fig. 13b, c) and some overlap between OA patients and non-OA/healthy subjects (Supplementary Fig. 13a, d). Using STRING-db[59], we identified an enrichment of the integrin cell surface interactions, ECM proteoglycans and focal adhesion in the healthy synovial fluid (Supplementary Fig. 9e). When analyzing the TAILS data, we identified an enrichment of regulation of receptor-mediated endocytosis, regulation of humoral immune response and positive regulation of cytokine production in healthy synovial fluid that is lost in OA synovial fluids, which suggests an effect on the N-termini processing as identified the TAILS data (Fig. 8g). We also identified an enrichment of STAT3, NFKB1, RELA and CEBPA gene regulation in both healthy and OA synovial fluids suggesting a role for these genes in OA (Fig. 8h). In the preTAILS data, we identified an enrichment of scavenging of heme from plasma and APOL1 complex A in OA patients but a loss of leukocyte cell-cell adhesion, phagosome, and NABA core matrisome (Fig. 8i). Moreover, we identified a significant increase of cleaved PRG4 in OA as compared to healthy subjects.

## Discussion

OA is a disease that impacts all the tissues within the joint leading to pain, disability, and eventual joint replacement[3,60]. Although the underlying mechanisms of OA progression are unclear, effective medications to delay joint replacement represent a significant unmet medical need. Here, we show that tryptase β cleavage of PRG4 resulted in the loss of lubricating potential and increased activation of the NF-κB pathway via TLR signaling (Fig. 9). These observations, together with PRG4's previously demonstrated ability to inhibit disease progression, suggest that inhibition of tryptase β activity could be a promising therapeutic strategy for OA. Synovial mast cells were previously demonstrated to increase the pathogenesis of inflammatory arthritis via the regulation of protease-activated receptor 2 (PAR-2)[61,62]. Recent evidence suggested that IgE-mediated MCs is a mediator of OA progression[29]. In a DMM model, treatment with imatinib mesylate, an inhibitor of multiple receptor tyrosine kinases, including c-kit, which is essential for mast cell growth, resulted in a reduction of cartilage degradation, osteophyte formation, and synovitis[29]. Therefore, we investigated if tryptase β, a key MC protease, could contribute to OA pathogenesis and impact the lubricating functions of PRG4.

In humans, PRG4 is found in high concentrations in the synovial fluid of joints (~400 μg/ml), and in lower concentrations (<10 μg/ml) in blood[11,52]. PRG4 reduces friction between load-bearing surfaces, and we demonstrated that tryptase, which was able to cleave PRG4 at physiological levels (Fig. 1), could diminish PRG4 lubricating abilities (Fig. 2). Using proteomics and N-terminomics approaches, we then demonstrated that tryptase β cleaves PRG4 at multiple sites including a cleavage site at position $^{1330}K\downarrow G^{1331}$, which was also identified when we

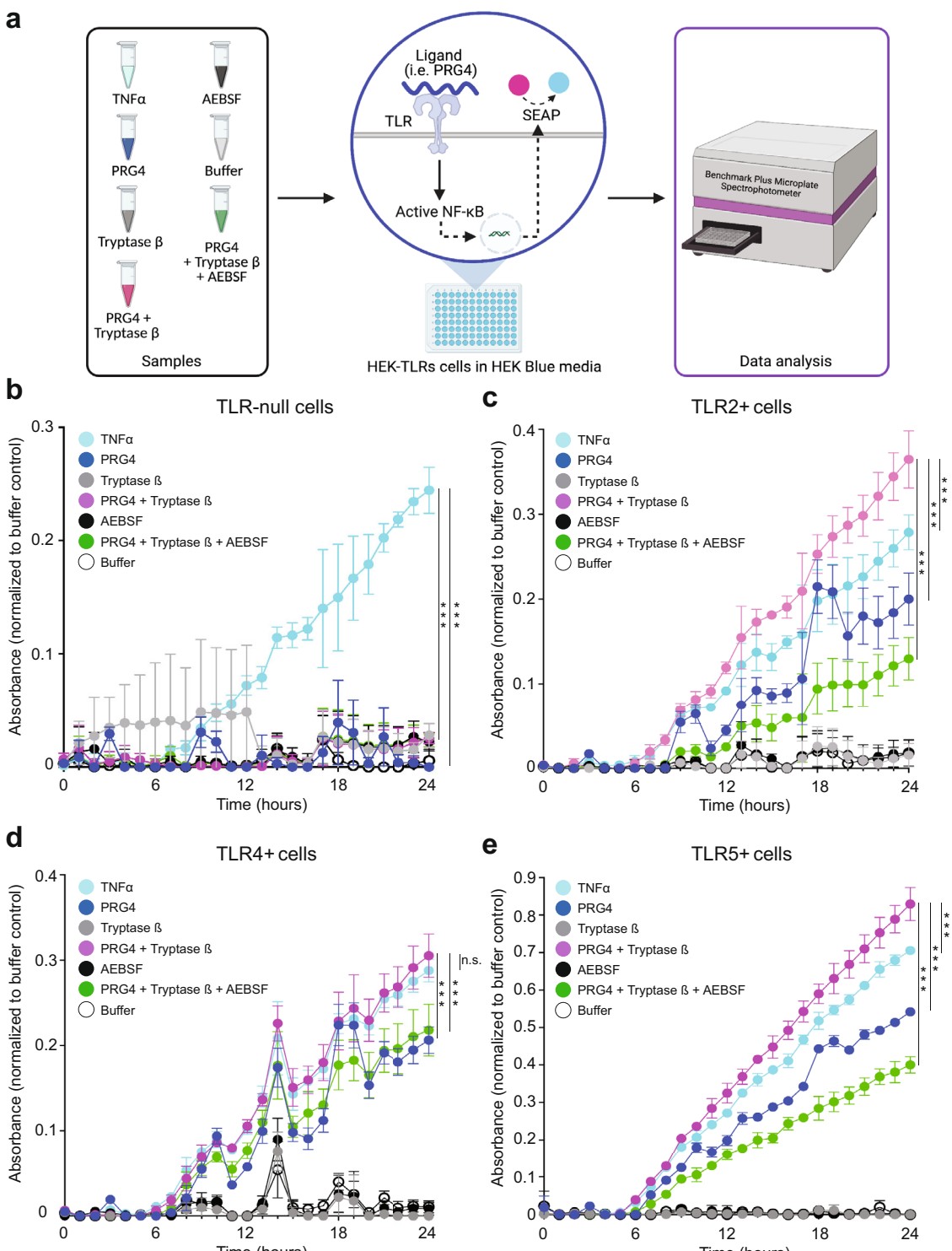

**Fig. 4 | NF-κB activation analysis in HEK-TLR reporter cells. a** Schematic of the experimental workflow of HEK-TLRS reported cells in HEK Blue media. Recombinant human (rh)-PRG4 (blue), rhPRG4 + tryptase β (magenta), tryptase β (dark grey), rhPRG4 + tryptase β + AEBSF (green), AEBSF (black), TNFα (pale blue), buffer (light grey) was added to HEK-TLRs cells and monitored over 24 h using the microplate reader for expression of the reporter gene (created with BioRender). All seven conditions were added in **b** *TLR*-null cells, **c)** *TLR2*⁻/⁻ cells, **d** *TLR4*⁻/⁻ cells and **e** *TLR5*⁻/⁻ cells. TNFα was used as a positive control and buffer was used as a negative control (*n* = 3 in **b**–**e**). Statistical analysis was determined at 24 h by a two-tailed unpaired Student's t test. For Fig. 4b–e, *p < 0.05, **, p < 0.01, ***, p < 0.001, and n.s., not significant were used. The error bars denote the error of the mean ± standard deviation as obtained from 3 independent experiments.

incubated recombinant tryptase in a complex ex vivo synovial fluid milieu (Fig. 7). We also found a cleavage site at ¹³⁰⁶K↓A¹³⁰⁷ (Fig. 1f) using ATOMS that was identified in OA patients' synovial fluid using N-terminomics (Fig. 8). Importantly, PRG4's glycosylation is not identical between healthy and OA patients[63]; and may differ to CHO cell O-glycosylated recombinant form[64]. Therefore, a simultaneous kinetic characterization of the glycosylation profiles and proteolysis of PRG4 in synovial fluids would be of interest to better understand its

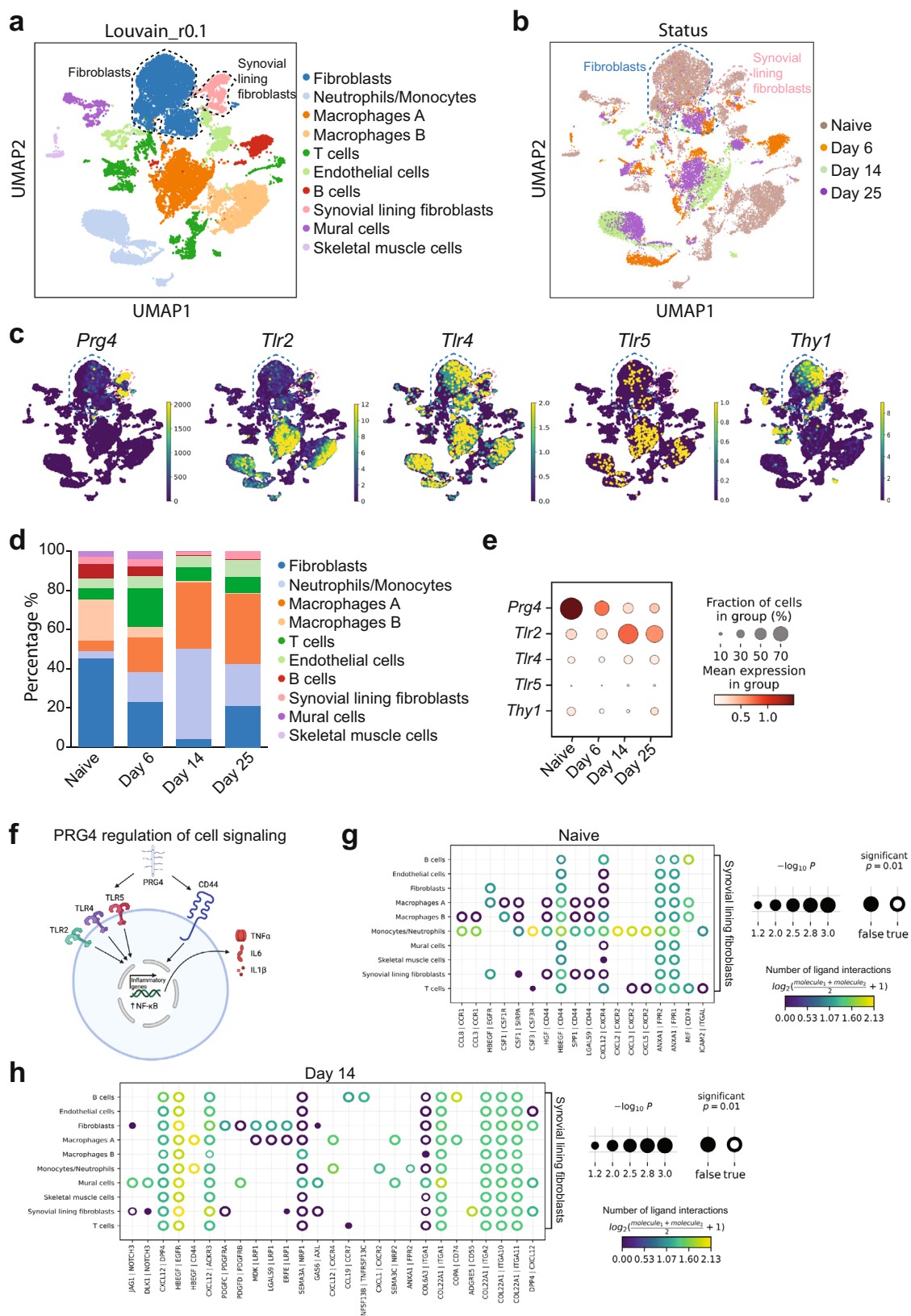

function in OA. We verified that PRG4 could activate NF-κB via TLR2, -4 and -5 and that cleaved PRG4 further increase NF-κB activation (Figs. 4 and 9). Interestingly, our data (Fig. 7j) demonstrated that adding tryptase β alone to human synovial fluid is sufficient to induce a change a gene expression in *NF-κB1* and *RELA*. In addition, we previously demonstrated that altered truncated PRG4 glycans can stimulate the

synoviocyte secretion of vascular endothelial growth factor a (VEGFA), IL8 and CCL3[65]. Recombinant PRG4 can induce NF-κB activation and requires tryptase β activity. Interestingly, other means of glycosylation, for example, O-GlcNAcylation, have also been demonstrated to contribute to NF-κB activation[66] and should be further investigated in OA. To validate these findings, we injected recombinant PRG4 into the

**Fig. 5 | Single-cell RNA-sequencing analysis of an inflammatory arthritis mouse model. a** Louvain clustering and cell annotation. Using Louvain clustering, we distinguished 10 cell clusters in the mice synovial biopsies with a resolution of 0.1. **b** Classification included cells based on the disease status, including naive (Day 0), Day 6, Day 14, and Day 25 of glucose-6-phosphate induction (GPI). **c** The expression pattern of *Prg4, Thy1, Tlr2, Tlr4,* and *Tlr5* in all clusters using UMAP. **d** Population distribution in percentages of scRNA-seq data of hind paw joint cells isolated at the indicated time points of GPI-induced arthritis. **e** The expression pattern of *Prg4, Thy1, Tlr2, Tlr4,* and *Tlr5* genes in the synovial fluid based on the status. **f** Schematic representation of PRG4 and its cell signaling (created with BioRender). **g** Analysis using the CellPhoneDB method[82] of the crosstalk between the synovial lining fibroblasts in the synovial fluid of naive (Day 0) and **h** Day 14 groups. For **g** and **h**, *p* values are computed from one-sided permutation test. Adjustments were made for multiple comparisons.

joints of rat with surgically induced OA (DMM model). Consistent with our previous studies, we observed that joints injected with recombinant PRG4 displayed delayed OA progression[8], yet we also observed that endogenous PRG4 colocalized with tryptase at 1-week and 2-weeks post-DMM (Fig. 3). Interestingly, the concentration of endogenous PRG4 in synovial fluid is inversely correlated to the level of inflammation[5,67]. Specifically, after a joint injury, inflammatory cytokines such as TNF-α and IL-1β increase dramatically, while PRG4 expression is near non-existent and does not return to normal levels until the inflammation has subsided[67]. Furthermore, pro-inflammatory mediators such as TNF-α and IL-1β act directly on mast cells to increase the expression and secretion of tryptase β[68,69]. The decrease in PRG4 levels has been associated with increased synovitis and cartilage damage[5,67,70], further suggesting that PRG4 plays a protective role within the joint and that factors present after injury, such as degranulating MCs contribute to OA pathogenesis. Importantly, MC numbers in synovial tissue correlate with structural damage in knee OA[71]. Wang et al.[29] treated mice subjected to the DMM mode with APC366, a small molecule inhibitor of serine proteases, including tryptase β, which resulted in a decrease of cartilage degradation, osteophyte formation and synovitis. Although APC366 is not a selective inhibitor of tryptase β, it serves as a rationale for inhibiting tryptase β in OA. Alternatively to MC, another interesting protease is Cathepsin G, present in neutrophils, which has been demonstrated to also degrade PRG4[64] and was identified in our shotgun proteomics data (Supplementary Table 3 and 7). A better characterization of the role of Cathepsin G in OA and its connection with MC and tryptase β is needed. Interestingly, the company Genentech® has produced highly selective allosteric anti-tryptase antibodies that inhibit human tryptase β activity[27,72]. Collectively, our data suggest that intra-articular treatment with a tryptase inhibitor together with PRG4, in OA or inflammatory state where PRG4 levels are diminished could help to maintain cartilage health and potentially slow the progression of OA. Our study may point to new therapeutic modalities that will help in slowing down the progression of OA.

## Methods

### Ethics statement
Our research complies with all relevant ethical regulations and the University of Calgary Research Ethics Board approved this study protocol. Our human studies were carried out in adherence to the principles of the Declaration of Helsinki. Informed consent to participate was obtained by written agreement. We obtained consent to publish information that identifies the age, sex, gender, and stage of disease. The study protocol was approved by the University of Calgary Research Ethics Board. All methods were carried out in accordance with the approved guidelines. Animal studies were carried out in accordance with the recommendations in the Canadian Council on Animal Care Guidelines. Animal protocols and surgical procedures in this study were approved by the University of Calgary Health Sciences Animal Care Committee.

### Tryptase β and PRG4 cleavage assays
PRG4 was previously generated[73] using Chinese hamster ovary (CHO) cells transfected with the PRG4 gene as generated by Lubris, LLC (Framingham, MA) in collaboration with Selexis SA (Geneva, Switzerland). Human recombinant beta tryptase (Promega corporation, WI, USA) was incubated with recombinant human PRG4 (Lubris Biopharma, MA, USA) (initial mass of PRG4 1 μg) in a protease to protein molar ratio of 1:10, 1:100 and 1:1000 in assay buffer containing 40 mM 4-(2- hydroxyethyl)-1-piperazineethanesulfonic acid (HEPES) and 0.12 M NaCl at pH-7.4[74], at 37 °C overnight. A broad-spectrum serine protease inhibitor, 4-(2aminoethyl) benzenesulfonyl fluoride hydrochloride (AEBSF) (Sigma-Aldrich, Oakville, ON, catalog # 30827-99-7) was used to inhibit tryptase β. PRG4 was incubated with tryptase at a ratio of 1:100 (tryptase: PRG4) in presence of increasing concentration of AEBSF (1, 10 and 100 μM) at 37 °C overnight. The stability of the PRG4 fragments generated by tryptase activity was tested in a time-dependent manner (5, 15, 30, 60, 180, 240 and 1,080 minutes) at protease to substrate ratio of 1:100 at 37 °C. PRG4 and tryptase β were deglycosylated at denaturing and non-denaturing conditions with protein deglycosylation mix II (PNGase F, to remove N-linked glycans, and an *Enteroccocus faecalis* O-glycosidase) (New England Biolabs, USA) as directed by the manufacturer before being incubated for 1 or 18 h. Tryptase cleavage products were resolved in 10% tris-glycine SDS-PAGE at 150 volt for 90 minutes and visualized by silver staining[75] where the SDS-PAGE gel was fixed in 40% methanol, 10% acetic acid for 15 minutes after electrophoresis, followed with sensitization by soaking in potassium ferricyanide and sodium thiosulfate for 5 minutes. Mixture of potassium ferricyanide and sodium thiosulfate was prepared by crushing crystals in a molecular weight ratio of 5:8, respectively. For each gel, one spatula of this mixture was resuspended in 25 ml $H_2O$. Then the gel was rinsed in $H_2O$ twice for 10 min until it turned from yellow to colorless. Next, the gel was impregnated in 12 mM silver nitrate ($AgNO_3$) solution for 15 min. The gel was quickly rinsed twice with $H_2O$ followed by a rinse with 2.9% sodium carbonate solution. The gel was then developed in 25 ml 2.9% sodium carbonate solution with 25 μl of 37% formaldehyde. Once proteins were visualized, the developing reaction was stopped by adding 5% acetic acid.

### Amino-Terminal Oriented Mass Spectrometry of Substrates (ATOMS) analysis of PRG4 cleavage by tryptase β
ATOMS employs isotopic labeling and quantitative tandem mass spectrometry to identify proteolytic cleavage sites[34]. Isotopic labeling was carried out as previously described[33,34]. 25 μg of proteins, either tryptase β digested PRG4 (in a protease to substrate ratio of 1:100 for 1 h and 18 h at 37 °C) or PRG4 alone, were reduced using a final concentration of 25 mM dithiothreitol (DTT) (Gold Biotechnology, St-Louis, MO) in 200 mM HEPES at 55 °C for 1 h. Samples were then alkylated to a final concentration of 60 mM iodoacetamide (IAA) (GE Healthcare, Mississauga, ON) for 20 min in the dark at room temperature, followed by a quenching reaction to a final concentration of 40 mM DTT for 25 min at room temperature. The generated N-termini and lysines were isotopically labeled: the tryptase and PRG4 sample was isotopically labeled with a final concentration of 20 mM deuterated heavy formaldehyde ($^{13}CD_2O$) (Cambridge Isotope Laboratories, Tewsbury, MA) and PRG4 without tryptase β sample was labeled with a final concentration of 20 mM light formaldehyde ($^{12}CD_2O$) (VWR Chemicals, Mississauga, ON) with the addition of 40 mM sodium cyanoborohydride (Sigma-Aldrich, Oakville, ON). The pH was then adjusted to 6.5 and incubated at 37 °C overnight. Following overnight incubation, samples were combined. After adjusting the pH to 8, mixed

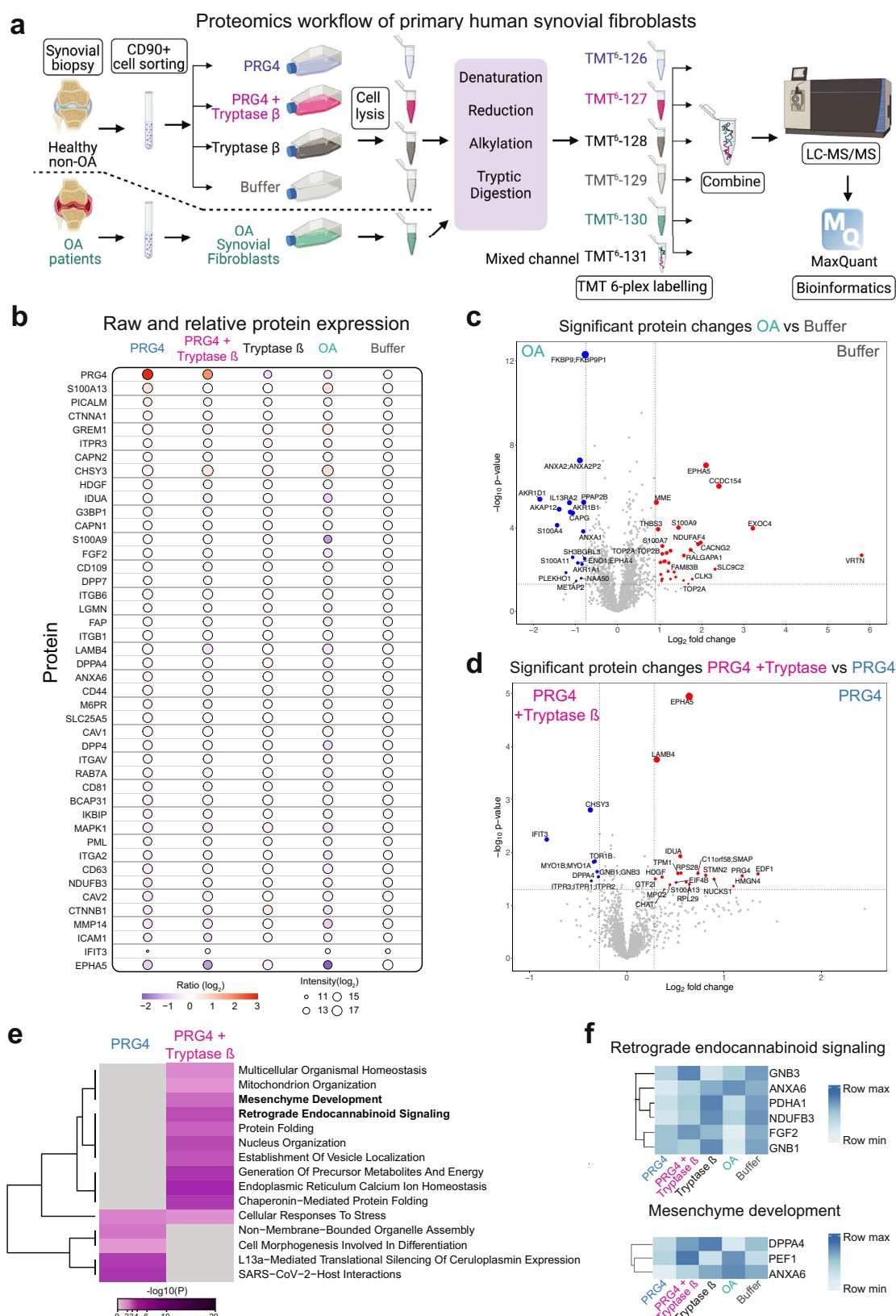

samples were digested with trypsin (Promega, Madison, WI) 1:10 protease to substrate molar ratio for 6 h at 37 °C. The samples were next acidified to a pH 2-3 with trifluoroacetic acid (TFA) and then desalted using Sep-Pak C18 columns (Waters Mississauga, ON). Sep-Pak columns were conditioned with 1 × 3 mL 90% methanol/0.1% TFA, 1 × 2 mL

0.1% formic acid. Each sample was loaded onto a column and washed with 1×3 mL 0.1% TFA/5% methanol. Peptides were eluted of the column with 1 × 1 mL 50% ACN/0.1% formic acid and lyophilized and submitted for liquid chromatography (LC)- tandem mass spectrometry (MS/MS) analysis to the Southern Alberta Mass Spectrometry core

**Fig. 6 | Proteomics analysis of primary human synovial fibroblasts from non-OA and OA patients. a** Workflow schematic of proteomics experimental design. Primary human synovial biopsies from non-OA and OA patients were isolated ($n = 3$ biological replicates, and 2 technical replicates). Primary synovial fibroblasts were sorted using CD90 (THY1). Non-OA primary fibroblasts were treated for 1 hour with PRG4 (blue), tryptase β + PRG4 1:100 enzyme to substrate ratio (magenta), tryptase β + buffer 1:100 (dark grey) or buffer (pale grey). Confluent OA primary fibroblasts were treated with buffer for 1 h (cyan). Next, cell were lysates and isotopically labeled with tandem mass tags (TMT) 6-plex and subjected to LC-MS/MS and analyzed using MaxQuant at 1% FDR. Graphic created using BioRender. **b** Dot plot analysis of raw and relative protein expression across the five conditions. The log2 ratio is shown as a gradient from blue to red and the log2 intensity of each protein is shown as the size of the circle. **c, d** Proteins identified by shotgun proteomics are represented as volcano plots. Log2 fold change using an interquartile boxplot

analysis is represented on the x-axis and -log10 $p$ value is represented on the y-axis. Two-sided analysis was performed, and it was adjusted for multiple comparisons. The complete list of proteins identified is shown in Supplementary Table 1. **e** Metascape[48] analysis of different pathways between PRG4 and PRG4 + tryptase β. Accumulative hypergeometric p-values and enrichment factors were calculated and used for filtering as performed as a two-sided analysis. Remaining significant terms were then hierarchically clustered into a tree based on Kappa-statistical similarities among their gene's memberships. Then, 0.3 kappa score was applied as the threshold to cast the tree into term clusters. **f** Heatmap of proteins and associated reactome pathways as determined by Metascape[48]. Interquartile box plot and -log10 adjusted p-value analysis was used for statistics. Data analysis was accomplished using the R software[94]. The plot was generated using the heatmap.2 function from the gplots package[95].

facility, University of Calgary, Canada. The LC-MS/MS data are analyzed using the database search MaxQuant software package v.1.6.0.1 at a peptide-spectrum match false discovery rate (FDR) of <0.01. Search parameter was specified for dimethylation of the N-termini and lysines as fixed modifications. This key feature ensures fully tryptic peptides, which canonically lack a dimethylated N-terminus and are ignored by the search engine, leading to data enriched for protease-generated peptides labeled both at their N-terminus and lysines if present.

### Friction/lubrication test

Lubricating ability of PRG4 and cleaved-PRG4 were analyzed in a tribology test with a glass-polydimethylsiloxane (PDMS) polymer interface using a ball-on-cylinder geometry on a MCR 302 rate-controlled rotational rheometer equipped with a tribology unit (Anton Paar, Graz, Austria) as previously described[38]. Velocity sweep analysis at different lubricating regimes was used to test the lubrication modes of the interface. For the ball-on-cylinder geometry, a borosilicate glass ball and 6 mm PDMS plugs were used. PDMS plugs were prepared by mixing silicone elastomer base and curing agent (Sylgard 184 Silicone Elastomer, Dow Corning Corporation, Midland, MI) at 10:1 ratio totaling a volume of 44 mL which was centrifuged at 300 rpm for 5 min to eliminate air bubbles before pouring into a petri dish. This mixture was then left covered overnight at room temperature. Three-cylinder-shaped plugs were placed into the sample holder in the tribology measuring cell after washing thoroughly with $H_2O$ followed by 70% ethanol and air-drying. One mL of sample was added to the sample holder, which was washed with $H_2O$ 3x after each sample. Samples concentration were 250 μg/mL for PRG4, 2.5 μg/mL for tryptase β and protease to substrate ratio for PRG4 processing with tryptase was 1:100 (2.5 μg/mL tryptase :250 μg/mL PRG4). Measuring shaft and glass ball were washed with $H_2O$ followed by 70% ethanol and dried in air before assembling. The tribology test sequence consisted of a 5 min incubation, followed by pre-conditioning, where the spinning velocity of the ball was increased from 0-1000 mm/s. Once a velocity of 1000 mm/s was reached, 26 measurements were recorded as the velocity was decreased to 0.01 mm/s. All tribology measurements were conducted and averaged for $n = 5$ trials. Measurements were conducted at a controlled temperature of 37 °C and a normal tribological force of $6.0 \pm 0.1$ N at the glass-PDMS interface. To access the lubricating boundary ability of each test lubricant, the kinetic coefficient of friction ($\mu_{kinetic}$) was measured in velocity curves.

### Tissue cytometry

For quantitative analysis of the histology data, the area of interest was acquired as digital greyscale images. Cells of a given phenotype were identified and quantitated using the TissueQuest v7.1 software (TissueGnostics), with cut-off values determined relative to the negative controls (non-stained and secondary alone controls). Gating and

quantification of single/double positive cells were undertaken using these thresholds.

### Single-cell RNA sequencing data acquisition

The raw single-cell RNA sequencing (scRNA-Seq) data of fluorescence-activated cell sorting (FACS)-sorted live synovial cells from naïve mice (two replicates) and mice on days 6, 14, or 25 of glucose-6-phosphate isomerase (GPI)-induced arthritis (one replicate per time point) were obtained from a previous analysis by Muench et al.[41]. The raw data from single-cell RNA sequencing is accessible in GEO (https://www.ncbi.nlm.nih.gov/geo/) under the accession number GSE184609 and the GPL24247 platform. Data analysis was carried out using the SCANPY pipeline[76]. For further validation, two other datasets were analyzed related to post-traumatic osteoarthritis (PTOA) progression and murine model of ACL injury. The accession numbers are GSE176308[77] and GSE211584[78].

### Single-cell RNA sequencing quality control and dimensionality reduction

We performed quality control to exclude low-quality and dead cells; we only included the cells that contained (1) >50 genes, (2) <30,000 counts, and (3) <20% of reads mapped to mitochondrial genes. Normalized expression was performed in the SCANPY pipeline using the normalize_total function or by estimating size factors for each cell to minimize bias within cell counts and enhance intercellular comparability of cell expression levels. Dimensionality reduction was conducted on the top 4,000 most highly variable genes (HVGs) to enable unsupervised grouping and cell-type identification by using principal component analysis (PCA). Then, Louvain community detection was embedded into a k-nearest-neighbor graph. Afterward, we created uniform manifold approximation (UMAP) embeddings to visualize this most relative neighbor graph using a minimal effective distance of 0.5, spread of 1.0[79]. We performed clustering with a resolution of 0.1 to identify subpopulations of synovial cells. The cell types were determined by the levels of expression of marker genes[80]. The markers for annotating individual cells were adopted from a combination of previous study[41] and The Human Protein Atlas website (https://www.proteinatlas.org/).

### Quality control, dimensionality reduction, clustering, and visualization

We applied quality control to exclude cells with poor quality. We only included cells that met the following criteria:

GSE176308[77]: (1) had more than 2,000 genes, (2) had less than 27,000 counts, and (3) had fewer than 20% of reads map to mitochondrial genes.

GSE211584[78]: (1) had more than 250 genes, (2) had less than 30,000 counts, and (3) had fewer than 20% of reads map to mitochondrial genes.

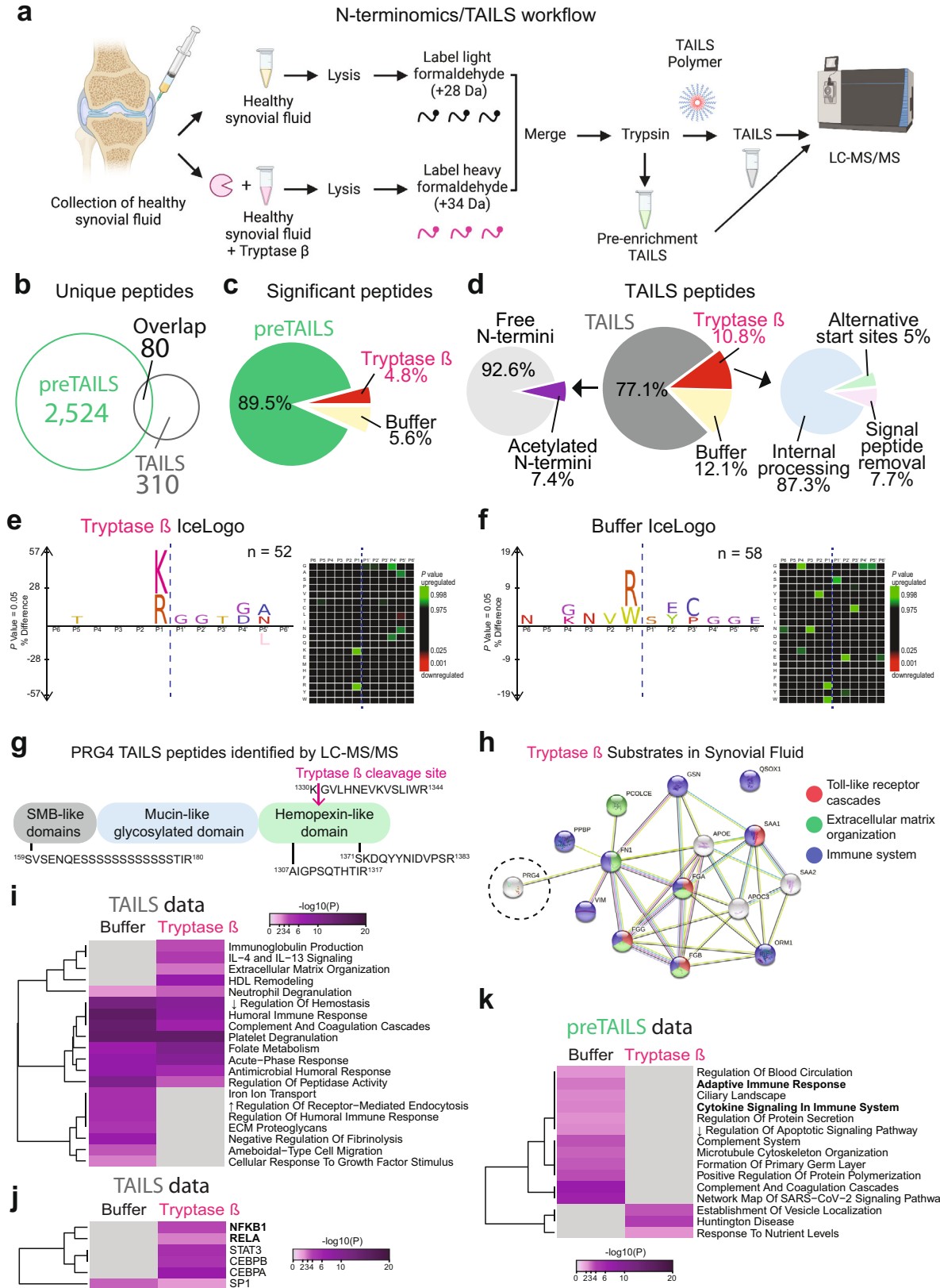

## Single-cell RNA sequencing cell-cell interaction analysis

SquidPy was applied to characterize cell-cell interaction[81], which provides an analytical method for storing, manipulating, and interactively visualizing single-cell RNA sequencing data. It re-implements the CellPhoneDB method[82] that is capable of handling a high number of interacting pairs (>100,000) and cluster combinations (>100).

## Patient inclusion and exclusion criteria

Healthy Group (*n* = 7): Criteria for control cadaveric donations were an age of 40 years or older, no history of arthritis, joint injury, or surgery (including visual inspection of the cartilage surfaces during recovery), no prescription anti-inflammatory medications, no co-morbidities (such as. diabetes/cancer), and availability within 4 h of death. OA

**Fig. 7 | Proteomics and N-terminomics/TAILS analyses of synovial fluids treated with buffer or tryptase β. a** Workflow of proteomics experimental design. Human synovial fluids from non-OA patients ($n = 7$) were incubated for 1 h with buffer or tryptase β at 1:100. Created using BioRender. **b** The numbers of unique and shared peptides between TAILS and preTAILS analysis. For a complete list, see Supplementary Tables 2–6. **c** The numbers of statistically changing peptides using an interquartile boxplot analysis between in the preTAILS samples. For a complete list, see Supplementary Table 2. **d** *Left*, Distribution of N-terminal peptides in the TAILS enrichment. *Middle*, statistically changing peptides using an interquartile boxplot analysis. *Right*, Distribution of post-translational peptide modifications as analyzed using TopFINDER[93]. For a complete list, see Supplementary Tables 5, 6. **e** *Left*, peptide sequence profiles of significantly elevated neo-N-terminal peptides in tryptase β-treated synovial fluids identified in the TAILS analysis using IceLogo[58]. **f** *Left*, peptide sequence profiles of significantly elevated neo-N-terminal peptides in buffer-treated synovial fluids identified in the TAILS analysis using IceLogo[58]. Significantly ($p < 0.05$) overrepresented amino acids are shown above, and

underrepresented residues are shown below the x-axis. *Right*, Cleavage sites identified tryptase β-treated synovial fluids are depicted as heat maps from P6 to P6′ residues. Green: Upregulated. Red: Downregulated. Statistical analysis was determined by a two-tailed unpaired Student's t test and was adjusted for multiple comparisons. **g)** All identified PRG4 peptides. The $^{1330}$K↓G$^{1331}$ N-termini was significantly elevated in the Tryptase-treated samples. **h** STRING-db[59] analysis of N-termini elevated in the Tryptase-treated samples from Supplementary Table 6. An enrichment was detected for Toll-like receptor (red), Extracellular matrix organization (green), and Immune system (blue). Metascape[48] **i** pathway enrichment and **j** TRRUST analysis of the TAILS data and **k)** Metascape[48] analysis of the preTAILS data of different pathways between buffer- and tryptase β-treated synovial fluids. Accumulative hypergeometric p-values and enrichment factors were calculated and used for filtering as performed as a two-sided analysis (for **i**, **j** and **k**). Remaining significant terms were hierarchically clustered into a tree based on Kappa-statistical similarities among their gene's memberships. Then, 0.3 kappa score was applied as the threshold to cast the tree into term clusters.

---

Group ($n = 3$): Criteria were an age of 40 years or older, OA diagnosed based on the American College of Rheumatology criteria with X-ray documentation, and no evidence of autoimmune disease or RA. Synovial biopsies. from the medial compartment were collected during routine arthroscopy. Only patients with an Outerbridge score of 3 or greater were selected for this study.

### N-terminomics/terminal isotopic labeling of substrates (TAILS) and shotgun proteomics of ex vivo human synovial fluid

Synovial fluid collected from healthy human knee joints ($n = 7$) were treated with tryptase (5 μg/1 mg of synovial fluid proteins) at 37 °C for 1 h equivalent of 1:100 tryptase β:proteome ratio. Also, synovial fluids from OA patient synovial fluids ($n = 3$) or healthy human knee joints ($n = 3$) were analyzed. After tryptase incubation, both treated and untreated synovial fluid or OA vs healthy synovial fluids were treated with 6 M guanidine HCl (pH 8.0) and subjected to a N-terminomics/TAILS[54,56,83–87] and shotgun proteomics workflow[88]. Samples were reduced with 5 mM DTT (Gold Biotechnology, St-Louis, MO) at 37 °C for 1 h and alkylated with 15 mM IAA (GE Healthcare, Mississauga, ON) in the dark at room temperature for 30 min followed by quenching with 15 mM DTT. The pH was adjusted to 6.5 before the samples were isotopically labeled with a final concentration of 40 mM deuterated heavy formaldehyde (tryptase treated or OA patients) and control samples (untreated or healthy subjects) with 40 mM light formaldehyde in presence of 40 mM sodium cyanoborohydride overnight at 37 °C. Next, samples were combined and were precipitated using acetone/methanol (8:1). The resulting pellet was resuspended in 1 M NaOH and the proteins were subjected to trypsin (Promega, Madison, WI) digestion overnight at 37 °C. For pre-enrichment TAILS (pre-TAILS)/shotgun proteomics, 10% of the trypsin-digested samples were collected and the pH was adjusted to 3 with 100% formic acid. The rest of the samples were adjusted to a pH of 6.5 and incubated with a 3-fold excess (w/w) of dendritic polyglycerol aldehyde polymer overnight at 37 °C[56,83]. Unbound peptides from the polymer-bound peptides were filtered out by centrifugal filter unit with 10-kDa cut-off membrane (Amicon Ultra, Millipore) at 10,000 g for 5 min. The flow-through was collected and the Amicon columns were washed with 100 mM Tris-HCl, ph 6.5. The pH of the samples was adjusted to 3 with 100% Formic acid. Both pre-TAILS and TAILS samples were then desalted using Sep-Pak C18 columns and lyophilized before submitting for LC-MS/MS analysis to the Southern Alberta Mass Spectrometry core facility, University of Calgary, Canada.

### Synovial membrane tissue digestion and fibroblast isolation

The intimal layer was dissected from the synovial biopsies ($n = 6$) and then cut into 5 mm² pieces. These were digested in 1 mg/ml type IV collagenase (Sigma) for 2 h at 37 °C while shaking. The cells were then washed with phosphate-buffered saline (PBS) and immediately

prepared for fluorescent activated cell sorting (FACS) on a BD FACS Aria Fusion (BD Biosciences, Mississauga, ON). The fibroblast marker was CD90 (Clone # 5E10, PE; catalog #555596), the macrophage marker was CD68 (clone # Y1/82 A, FITC; catalog #562117), and a cell viability marker was FVS510 (BV510; catalog # 564406) (all BD Biosciences, Mississauga, ON). UltraComp eBeads (eBioscience, Mississauga, ON) individually stained with each single colour as well as unstained cells were used as compensation controls. All antibodies were used at a 1:1000 dilution. Macrophages (CD68$^+$) as well as the dead cells (FVS510$^+$) were excluded. The remaining cells were sorted based on CD90$^+$ expression. The sorting was undertaken using a 100 μM sort nozzle and low flow rate (45% of system maximum) to reduce the pressure on the cells.

### Quantitative shotgun proteomics using tandem mass tags-6 (TMT-6) labeling

Synovial membrane fibroblasts collected from healthy patients ($n = 6$) and osteoarthritis patients ($n = 6$) were treated with PRG4 (50 μg/mL), tryptase β processed PRG4 (1:100), tryptase β alone (0.5 μg/mL) or buffer for 48 h. Cells were treated once they reached 60% confluency in a 6-well plate and incubated at 37 °C and 5% $CO_2$. After an incubation of 48 h, cells were removed from the flask and lysed in 400 μL lysis buffer (1% SDS, 100 mM ammonium bicarbonate, 0.2 mM EDTA and protease inhibitor tablets (cOmplete™ Protease Inhibitor Cocktail, Roche)). Samples were then sonicated three times for 5 seconds on ice. Then the lysates were centrifuged at 8000 g for 10 min and supernatants were collected carefully followed by protein quantification using a BCA kit. One hundred μg cellular proteins were isolated for a TMT-6 plex shotgun proteomics[89,90]. TMT 6-plex labeling was carried out following the manufacturer's protocol (Thermo Fischer Scientific, Mississauga, ON). One hundred μg lysates were topped up to 400 μL with 100 mM triethyl ammonium bicarbonate (TEAB) and reduced with 10 mM Tris(2-carboxyethyl) phosphine hydrochloride (Thermo Fisher Scientific, Mississauga, ON) at 55 °C for 1 h followed by alkylation with 15 mM IAA for 25 min in the dark at room temperature. Proteins were precipitated by adding 600 μL of prechilled (−20 °C) acetone and left at −20 °C overnight. Samples were centrifuged at 8000 g for 10 min before resuspension in 100 μL of 50 mM TEAB. Proteins were next digested with trypsin (Promega, Madison, WI) at 1:10 ratio at 37 °C overnight. For TMT 6-plex labeling, TMT reagents were first equilibrated at room temperature, and then 0.8 mg of the reagents were resuspended in 41 μL 100% acetonitrile (ACN). Samples were spun down at 380 g for 10 s before adding the TMT reagents and then incubated at room temperature for 1 h. The labeling reaction was quenched by adding 8 μL of 5% hydroxylamine and incubated for 15 min at room temperature. Peptides were combined before being acidified to pH 3 with TFA, were then desalted using Sep-Pak C18 columns and lyophilized before submitting for LC-MS/MS analysis at the

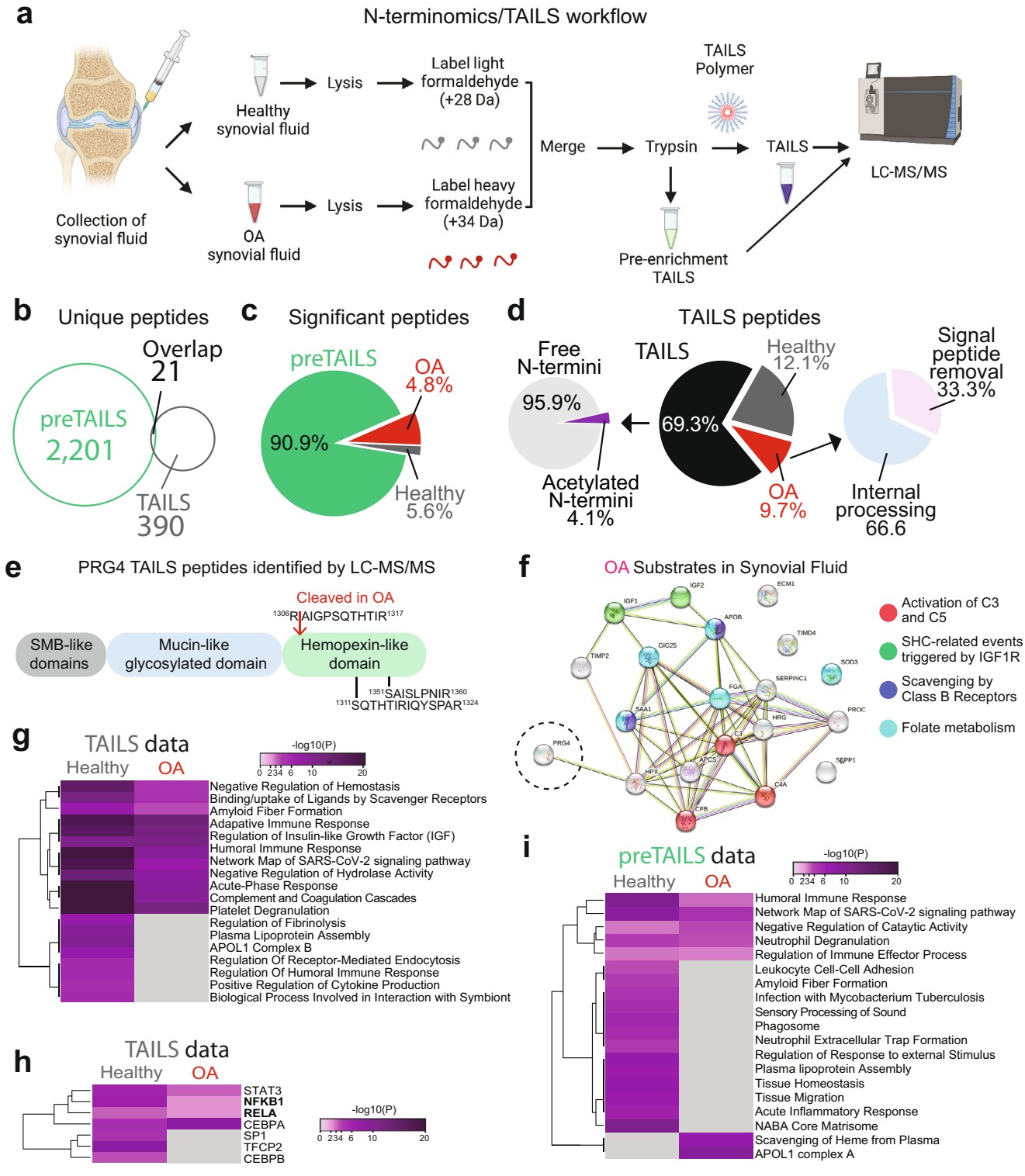

**a** N-terminomics/TAILS workflow

**b** Unique peptides

**c** Significant peptides

**d** TAILS peptides

**e** PRG4 TAILS peptides identified by LC-MS/MS

**f** OA Substrates in Synovial Fluid

**g** TAILS data

**h** TAILS data

**i** preTAILS data

Southern Alberta Mass Spectrometry core facility, University of Calgary, Canada.

**High-performance liquid chromatography (HPLC) and mass spectrometry**

All liquid chromatography and mass spectrometry experiment were carried out by the Southern Alberta Mass Spectrometry core facility at the University of Calgary, Canada. Analysis was performed on an Orbitrap Fusion Lumos Tribrid mass spectrometer (Thermo Fisher Scientific, Mississauga, ON) operated with Xcalibur (version 4.0.21.10)

and coupled to a Thermo Scientific Easy-nLC (nanoflow Liquid Chromatography) 1,200 system. Tryptic peptides (2 μg) were loaded onto a C18 trap (75 μm × 2 cm; Acclaim PepMap 100, P/N 164946; Thermo Fisher Scientific) at a flow rate of 2 μL/min of solvent A (0.1% formic acid and 3% acetonitrile in LC-mass spectrometry grade water). Peptides were eluted using a 120 min gradient from 5 to 40% (5% to 28% in 105 min followed by an increase to 40% B in 15 min) of solvent B (0.1% formic acid in 80% LC-mass spectrometry grade acetonitrile) at a flow rate of 0.3% μL/min and separated on a C18 analytical column (75 μm × 50 cm; PepMap RSLC C18; P/N ES803; Thermo Fisher

**Fig. 8 | Proteomics and N-terminomics/TAILS analyses of synovial fluids human OA and non-OA patients. a** Workflow schematic of proteomics experiments. Human synovial fluids from healthy/non-OA patients ($n = 3$) were compared to OA patients ($n = 3$). Graphic created using BioRender. **b** The numbers of unique and shared peptides between TAILS and preTAILS analysis. For a complete list, see Supplementary Tables 7–11. **c** The numbers of statistically changing peptides using an interquartile boxplot analysis between in the preTAILS samples. For a complete list, see Supplementary Table 7. **d** *Left*, Distribution of N-terminal peptides in the TAILS enrichment. *Middle*, The numbers of statistically changing peptides using an interquartile boxplot analysis between in the TAILS samples. *Right*, Distribution of post-translational peptide modifications, as analyzed using TopFINDER[93]. For a complete list of N-termini identified, see Supplementary Tables 9–11. **e** All identified PRG4 peptides. The $^{1306}R{\downarrow}A^{1307}$ N-termini was significantly elevated in the OA

samples as compared to non-OA/healthy synovial fluid. **f** STRING-db[59] analysis of all N-termini elevated in the OA samples from Supplementary Table 11. An enrichment was detected for Activation of C3 and C5 (red), SHC-related events triggered by IGF1R (green), Scavenging by Class B Receptors (blue), and Folate metabolism (pale blue). Metascape[48] **g** pathway enrichment and **h** TRRUST analysis of the TAILS data and **i** Metascape[48] analysis of the preTAILS data of different pathways between non-OA/healthy and OA synovial fluids. Accumulative hypergeometric p-values and enrichment factors were calculated and used for filtering as performed as a two-sided analysis (for **g**, **h** and **i**). Remaining significant terms were then hierarchically clustered into a tree based on Kappa-statistical similarities among their gene's memberships. Then, 0.3 kappa score was applied as the threshold to cast the tree into term clusters.

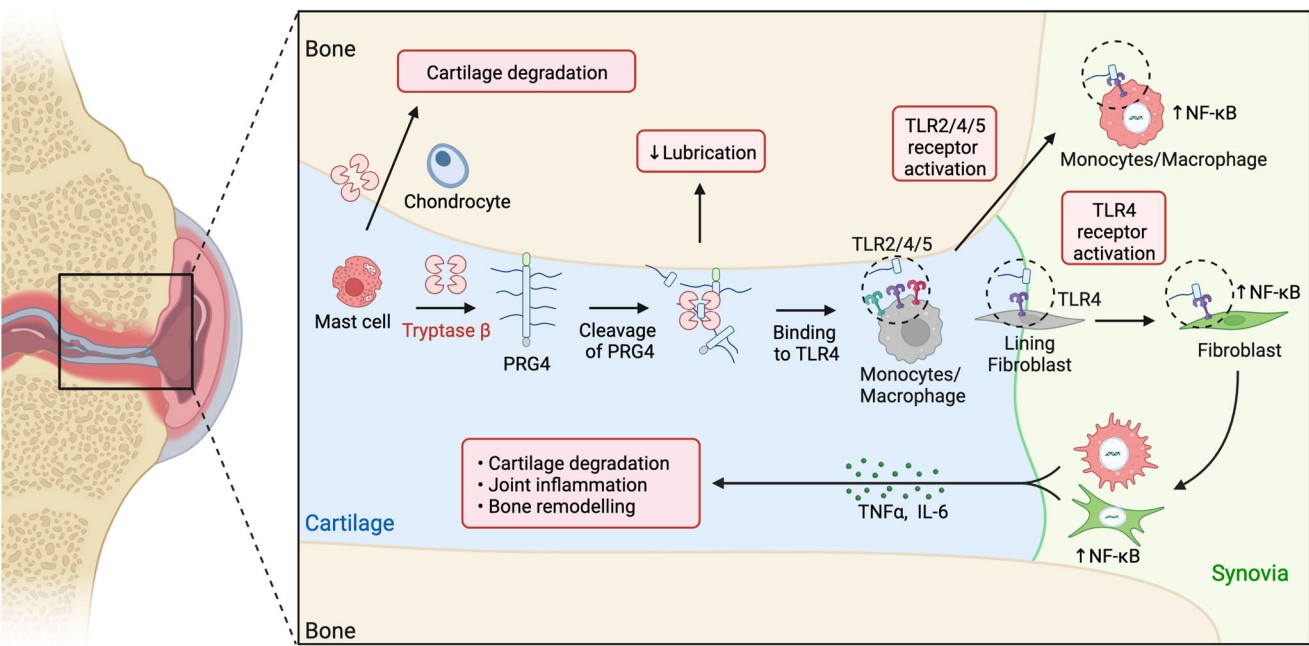

**Fig. 9 | Mechanisms of PRG4 processing by Tryptase β.** Workflow schematic of tryptase β processing of PRG4 resulting in TLR activation and NF-κB activation in human synovial fluids in OA patients. Graphic created using BioRender.

Scientific). Peptides were then eletrosprayed using 2.3 kV into the ion transfer tube (300 °C) of the Orbitrap Lumos operating in positive mode. The Orbitrap first performed a full mass spectrometry scan at a resolution of 120, 000 FWHM to detect the precursor ion having a mass-to-charge ratio (m/z) between 375 and 1575 and a +2 to +4 charge. The Orbitrap AGC (Auto Gain Control) and the maximum injection time were set at $4 \times 10^5$ and 50 ms, respectively. The Orbitrap was operated using the top speed mode with a 3 s cycle time for precursor selection. The most intense precursor ions presenting a peptidic isotopic profile and having an intensity threshold of at least $2 \times 10^4$ were isolated using the quadrupole (isolation window of m/z 0.7) and fragmented with HCD (38% collision energy) in the ion routing Multipole. The fragment ions (MS2) were analyzed in the Orbitrap at a resolution of 15,000. The AGC, the maximum injection time and the first mass were set at $1 \times 10^5$, 105 ms, and 100 ms, respectively. Dynamic exclusion was enabled for 45 s to avoid of the acquisition of the same precursor ion having a similar m/z (±10 ppm).

### Proteomic data and bioinformatic analysis
Spectral data were matched to peptide sequences in the human Uni-Prot protein database using the MaxQuant software package v.1.6.0.1, peptide-spectrum match false discovery rate (FDR) of <0.01 for the shotgun proteomics data and <0.05 for the N-terminomics/TAILS data. Search parameters included a mass tolerance of 20 p.p.m. for the

parent ion, 0.05 Da for the fragment ion, carbamidomethylation of cysteine residues (+57.021464), variable N-terminal modification by acetylation (+42.010565 Da), and variable methionine oxidation (+15.994915 Da). For the shotgun proteomics data, cleavage site specificity was set to Trypsin/P (search for free N-terminus and only for lysines), with up to two missed cleavages allowed. The files evidence.txt and proteinGroups.txt were analyzed by MSstatsTMT (v2.4.0)[43] using R software (v4.2.0) (R Core Team (2022). R: A language and environment for statistical computing. R Foundation for Statistical Computing, Vienna, Austria. URL https://www.R-project.org/) for the statistical analysis. For the N-terminomics/TAILS data, the cleavage site specificity was set to semi-ArgC (search for free N-terminus) for the TAILS data and was set to ArgC for the preTAILS data, with up to two missed cleavages allowed. Significant outlier cut-off values were determined after log(2) transformation by boxplot-and-whiskers analysis using the BoxPlotR tool[91]. Database searches were limited to a maximal length of 40 residues per peptide. Peptide sequences matching reverse or contaminant entries were removed.

### Reactome pathway analysis
To identify interconnectivity among proteins, the STRING-db (Search Tool for the Retrieval of Interacting Genes) database was used to identify interconnectivity among proteins. The protein-protein interactions are encoded into networks in the STRING.v11[59] database

(https://string-db.org). Metascape[48] (https://metascape.org) analysis was used to identify changes in functional enrichment, interactome analysis, and gene annotation. Our data were analyzed using *Homo sapiens* as our model organism at a false discovery rate of 1%.

## Heatmaps of cleavage sites, TopFIND and TopFINDer analysis

WebPICS was used using the website http://clipserve.clip.ubc.ca/pics. TopFIND[92] and TopFINDer[93] analyses were performed using the Web site http://clipserve.clip.ubc.ca/topfind/. Uniprot (https://www.uniprot.org/) and MEROPS (https://www.ebi.ac.uk/merops/index.shtml) were used to interpret the data.

## TLR activation assay in HEK cells

Human *TLRnull*, TLR2+, TLR4+ and TLR5+ cell lines (Invivogen, San Diego, CA) were cultured as directed by the manufacturer. For the activation assay, 20 μL of either a positive control for TLR, TNFα (100 ng/mL) or PRG4 (50 μg/mL) were loaded onto a 96-well plate. Both PRG4 and tryptase β processed PRG4 (1:100 in the presence or absence of AEBSF (10 μM) at 37 °C overnight) were added to the cells. After 3x washing in PBS, cells were diluted to a concentration of $2 \times 10^5$ cell/mL in HEK-blue detection medium (Invivogen, San Diego, CA) whereupon 180 μL of the cell suspension was immediately added to the sample dilutions and incubated at 37 °C, 5% $CO_2$ over 24 h. Change in HEK media color was measured at 630 nm every hour using Benchmark Plus Microplate Spectrophotometer (BioRad, CA, USA).

## Destabilization of the Medial Meniscus (DMM) injury model

To induce standardized joint injury in knees ($n = 9$), each rat (8 female and 5 male 10-week-old Lewis rats were used in total) was anesthetized with isoflurane delivered in oxygen and underwent medial para-patellar arthrotomy under a surgical microscope. At the time of surgery, males weighed -250 g +/− 50 g and females weighed 175 g +/− 20 g. The joint was opened by separating the medial margin of the quadriceps from the muscles of the medial compartment and laterally dislocating the patella. the fat pad over the cranial horn of the medial meniscus was retracted and the medial meniscotibial ligament was cut to destabilize the medial meniscus. The joint capsule was closed with a 4-0 nylon suture and the skin by the application of tissue adhesive. Control groups underwent sham surgery ($N = 9$). One week after the DMM surgery, each rat received intra-articular PRG4 injection at a dose of a dose 200 μg/kg in 10 μL sterile PBS saline. Control animals received an equal volume of sterile saline.

## Multiplex analysis of cytokines and chemokines

Rat serum was collected from the tail vein at 4-week post-DMM injury. Samples were stored at −80 °C and analyzed together to minimize inter-experimental variation. We used Luminex xMAP technology for multiplexed quantification of rat cytokines, chemokines, and growth factors. The multiplexing analysis was performed using the Luminex™ 200 system (Luminex, Austin, TX, USA) by Eve Technologies Corp. (Calgary, Alberta). Seven markers were simultaneously measured in the samples using Eve Technologies' Rat Cytokine Plex Discovery Assay® (MilliporeSigma, Burlington, Massachusetts, USA) according to the manufacturer's protocol. The 7-plex consisted of G-CSF, GM-CSF, IFNγ, IL1β, TNFα CCL2, and IL6. Assay sensitivities of these markers range from 0.3 to 30.7 pg/mL for the 7-plex. All cytokines and chemokines present were quantified in pg/mL and analyzed in duplicate alongside a reference sample and compared to an internal standard curve.

## Immunohistochemistry

Knee joints were collected from animals sacrificed at 1, 2, 3 and 4-weeks post-DMM surgery and 3-weeks after PRG4 injection (corresponding to 4-weeks after DMM surgery) and fixed in 4% normal buffered formalin (Sigma, St. Louis, MO). Samples were decalcified in EDTA for 3 weeks and embedded in paraffin (VWR, Radnor, PA). Ten μm thick, longitudinal serial sections were collected using microtome (Leica RM 2165, Leica Biosystems, ON, Canada). These sections were deparaffinized in CitraSolv (Fisher Scientific, Fairlawn, NJ) and rehydrated through a series of graded ethanol to distilled water steps. Antigen retrieval in 10 mM sodium citrate at pH 6.0 (Fisher Scientific, Mississauga, ON) for 1 h followed by blocking in 1:500 rat serum in TRIS-buffered saline in 0.1% Tween 20 (TBST) for 1 h were performed prior to going through sequential TBST wash and primary antibody application steps. Primary antibody for PRG4; mucin domain antibody 9G3 (Millipore, MABT401) bound to Dylight 630 (AbCam Millipore, ab201803) and C-terminal antibody, LPN (Invitrogen, PA3-118) bound to Dylight 488 (AbCam Millipore, ab201799) or beta-tryptase (Biolegend, 369402) bound to Dylight 550 (AbCam Millipore, ab201800) and the nucleic acid stain DAPI (Biotium, 23004) were applied to sections. Antibody dilution was performed as per manufacturer's recommendations. After antibody staining, sections were mounted using Everbright™ Mounting Medium (Biotium, 23004) and cover slipped.

## Statistical analysis

For the N-terminomics analysis, an interquartile boxplot analysis was applied to determine the differential enrichment/upregulation of proteins[91]. The proteomics data analysis was performed using files evidence.txt and proteinGroups.txt as analyzed by MSstatsTMT (v2.4.0)[43] using R software (v4.2.0) (R Core Team (2022). R: A language and environment for statistical computing. R Foundation for Statistical Computing, Vienna, Austria. URL https://www.R-project.org/). The FDR was generated by the bioinformatic tools used for pathway analysis. The R software, GraphPad Prism version 9, and Microsoft Excel were used for the statistical analysis. The Student's t-test and ANOVA determined significance between treatments using the Prism software. A $p$-value <0.05 was considered statistically significant.

## Reporting summary

Further information on research design is available in the Nature Portfolio Reporting Summary linked to this article.

## Data availability

All data generated or analysed during this study are included in this published article (and its supplementary information files). Proteomics RAW data were deposited to ProteomeXchange via the Proteomics Identification Database (PRIDE) under accession number PXD037040. All data about identified and quantified peptides and proteins are included in the Supplementary Tables 1–11. Gene Expression Omnibus accession numbers: GSE184609, GSE176308, and GSE211584. Source data are provided with this paper.

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

## Acknowledgements

We thank Laurent Brechemencher and the Southern Alberta Mass Spectrometry (SAMS) core facility for proteomic analysis. This study was supported by the STARS award by the Arthritis Society of Canada (R.K. and An.Du), the Canadian Institutes of Health Research (CIHR) (grant number 449589) (An.Du) and the Natural Science and Engineering Research Council of Canada (NSERC) (grant number DGECR-2019-00112) (An.Du). The Arthritis Society of Canada project grant (R.K. and An.Du).

## Author contributions

Conceptualization N.D., T.A.S., R.K., An.Du; methodology N.D., L.G.N.d., Af.De., K.M. T.A.S., R.K., An.Du; validation N.D., L.G.N.d., Af.De. T.A.S., R.K., An.Du; resources N.D., L.G.N.d., Af.De., P.S. G.D.J., C.S. T.A.S., R.K., An.Du; writing—review and editing N.D., L.G.N.d., Af.De., P.S. D.Y., A.R. G.D.J., C.S. T.A.S., R.K., An.Du; supervision, T.A.S., R.K. An.Du; funding acquisition T.A.S., R.K. An.Du.

## Competing interests

T.A.S. and G.D.J. have authored patents on rhPRG4 and hold equity in Lubris LLC, MA, USA. T.A.S. is also a paid consultant for Lubris LLC, MA, USA. R.K. also has authored patents on rhPRG4. All other authors have nothing to disclose.
