## [Peer Review File · Nature Communications]

REVIEWER COMMENTS

Reviewer #1 (Remarks to the Author):

In this manuscript by Das and colleagues showed that PRG4 can be cleaved by tryptase β and this cleavage correlates with increased NF- κ B activation. Using a multi-omics approach (proteomics and N-terminomics), they show that PRG4 is cleaved at multiple locations and that the proteome is altered following addition of tryptase β to CD90+ cells isolated from healthy synovial biopsies. In the TLR activation assay, there is clear activation in cells following treatment with tryptase β and this activation can be reduced following inhibition with a broad-spectrum serine protease inhibitor. Overall, these studies show that tryptase β may play a role in the ability of PRG4 to lubricate the joint.

The figures in this manuscript are very clear and I liked the cartoon images in panel a of several figures that show how the experiment was set up. Nice work!!

I have some points that I would like to see address before acceptance of this manuscript:

This study has extensive data showing that tryptase β is capable of cleaving PRG4 and that this cleavage results in a reduction of lubrication. However, there is not clear evidence that the same processing of PRG4 occurs in the joint of OA animals or patients. In Figure 1, cleavage of PRG4 yields a 50kDa product that is reasonably stable for up to 4 hours. Does this same product accumulate in the joints of the DMM model. The authors have two antibodies, one that binds to the mucin-like domain (M-PRG4 Ab) and one that binds to the hemopexin-like domain (C-PRG4 Ab). Can these antibodies be used to that the same PRG4 products are generated in-vitro and in vivo? This would add greater strength to the hypothesis that a loss of lubrication is directly due to tryptase activity in the joint.

In the TAILS experiment, there is at least one other protease present in synovial fluid that is capable of cleaving after Arg residues (Figure 7f). Can you identify this protease? It is possible that this enzyme will also be able to cleave PRG4. This would be a key discovery as there may be more than one protease in the joint that is capable of decreasing lubrication via cleavage of PRG4.

In Figure 2b, addition of Tryptase or Tryptase+ AEBSF cause and increase in the Coefficient of Friction. However, there are no controls showing that the addition of any protein would yield the same change in lubrication as is seen from PRG4+Tryptase+AEBSF. I would like to see what the addition of AEBSF alone would result in. Also, perhaps adding BSA (at the same conc as Tryptase) would be useful control. In Figure 2c, could earlier timepoints (e.g. 10 min and 30 min) be included to show that the coefficient of friction change is also time-dependent before 1 h.

Minor points:

1. In the introduction, there is a word missing in the sentence “a central mucin-like domain flanked by globular _____”

2. In the introduction, the following sentence does not make sense. “There is growing evidence that MCs play a key role in knee OA and by depleting mast cells”. Do mast cells deplete mast cells??

Reviewer #2 (Remarks to the Author):

Unraveling molecular mechanisms of Osteoarthritis (OA) initiation and progression has been a long sought after task. Human OA is a common but yet an extremely heterogeneous disease involving infiltration of diverse inflammatory cells, loss of boundary lubrication and degradation of articular cartilage. Integrating, biochemical and omics analyses, Das et al identified mast cell tryptase β as a modulator of joint lubrication in OA which is mediated by the cleavage of Proteoglycan 4 (PRG4) known to an important protein components for cartilage integrity. The authors monitored the degradation of PRG4 by tryptase β in vitro and characterized this process by MS using MDD model of OA in rats, and human synovial samples. OA severity could be associated with degraded PRG4 which activates NF-kB expression in cells overexpressing TLRs. The results indicate that maintaining intact PRG4 levels are key for cartilage integrity and lubrication. Overall, the reported experiments described by the authors are logic, clear, and well interpreted. The importance of PRG4 and its degraded fragments as an inflammatory signaling molecules was recently identified. Nevertheless, the identification of mast cell derived tryptase β as key enzyme in this process is an important new discovery. Thus inhibition of tryptase β enzymatic activity and maintaining normal physiological levels of PRG4, even locally, could be a promising therapeutic strategy for OA as suggested by the authors.

Nevertheless, this manuscript will benefit from the following suggested experiments and revisions:

1. In vivo inhibition of tryptase β , possibly by using the anti-tryptase inhibitory antibody, may be performed in DMM model to further evaluate it as key main factor in OA progression. Alternatively, treatment of DMM model with pan tryptase β inhibitor + PRG4 should be performed to validate the therapeutic potential of this treatment suggested by the authors.
2. The authors should specify the expression system of the recombinant proteins and their glycosylation profiles compared to the physiological proteins. Different glycosylation patterns may explain the differences in the cleavage sites identified in recombinant vs ex vivo PRG4. Are the fragments remained glycosylated. Is glycosylation responsible for NF-kB ? The authors should elaborate on this points in text.
3. Was tryptase β deglycosylated prior to treatment of synovial fluids? In Materials and methods, Fig 1b, d. Fig. 2b: need to specify the initial concentrations of PRG4.
4. Include image analysis in Fig.3 to show decrease/increase of PRG4 for both domains. In addition, statistical significance based on chosen analysis should be presented.

Reviewer #3 (Remarks to the Author):

This study by Das and colleagues examined the cleavage of proteoglycan-4 (PRG4) by beta-tryptase as it pertains to osteoarthritis. Specific cleavage sites and resulting products of proteolysis were identified. The effect of tryptase-induced PRG4 cleavage resulted in a reduction in the lubricating and joint protective properties of the glycoprotein. Combined inhibition of tryptase activity with PRG4 rescued the beneficial effects of PRG4. This is an interesting study which has been well executed and highlights a potential new treatment strategy for osteoarthritis.

1. Inhibition of tryptase activity with AEBSF improves the lubrication coefficient of PRG4 (Fig. 2B); however, it does not recover to PRG4 alone levels. The authors should discuss what other factors could be cleaving PRG4 to limit its lubricating properties.
2. In Fig. 3: does the “uninjured control” refer to sham-operated animals? If so please change. If not please add in a sham group. It would be nice to have a semi-quantification of the expression levels between the different groups (e.g. area of immunoreactivity).
3. The authors should demonstrate that inhibition of tryptase activity improves the protective effect of PRG4 on joint damage (i.e. OARSI scores).
4. The study would benefit greatly from some sort of functional read-out (e.g. improved gait, reduced inflammation in vivo).
5. When examining the expression of PRG4 in synovial fibroblasts, why did the experimenters change to the GPI model of inflammatory joint disease? Shouldn't this be carried out in the DMM model for consistency?
6. The sex, strain, and weight range of animals used in the study must be described.
7. Add page numbers for ease of review.

Here are our point-by-point answers to the reviewers:

Reviewer #1:

In this manuscript by Das and colleagues showed that PRG4 can be cleaved by tryptase β and this cleavage correlates with increased NF- κ B activation. Using a multi-omics approach (proteomics and N-terminomics), they show that PRG4 is cleaved at multiple locations and that the proteome is altered following addition of tryptase β to CD90+ cells isolated from healthy synovial biopsies. In the TLR activation assay, there is clear activation in cells following treatment with tryptase β and this activation can be reduced following inhibition with a broad-spectrum serine protease inhibitor. Overall, these studies show that tryptase β may play a role in the ability of PRG4 to lubricate the joint. The figures in this manuscript are very clear and I liked the cartoon images in panel a of several figures that show how the experiment was set up. Nice work!!

Thank you for your positive comment.

I have some points that I would like to see address before acceptance of this manuscript: This study has extensive data showing that tryptase β is capable of cleaving PRG4 and that this cleavage results in a reduction of lubrication. However, there is not clear evidence that the same processing of PRG4 occurs in the joint of OA animals or patients.

Following the reviewer's suggestion, we added a new analysis where we performed N-terminomics/TAILS and shotgun proteomics on the synovial fluid of OA patients and non-OA/healthy patients, as shown in **NEW Figure 8a-i**. We identified a significant increase of processed PRG4 in OA patients as compared to non-OA patients. Importantly, we found additional tryptase substrates, which is interesting. Characterizing the timing of when PRG4 is cleaved (early OA vs established OA) and at what sites as we identified multiple cleavage sites of PRG4 by beta-tryptase (**Figure 1e-g**) is still needed. However, we feel that these are complex questions that will warrant our effort in a subsequent publication.

In Figure 1, cleavage of PRG4 yields a 50kDa product that is reasonably stable for up to 4 hours. Does this same product accumulate in the joints of the DMM model. The authors have two antibodies, one that binds to the mucin-like domain (M-PRG4 Ab) and one that binds to the hemopexin-like domain (C-PRG4 Ab). Can these antibodies be used to that the same PRG4 products are generated in-vitro and in vivo? This would add greater strength to the hypothesis that a loss of lubrication is directly due to tryptase activity in the joint.

As we used human proteins in **Figure 1**, we thought that a better way to address this question was to continue using human samples. We have added a confirmation that PRG4 is cleaved in OA synovial fluids as shown in **NEW figure 8e**. In synovial fluid, similar to serum or plasma, there is an overabundance of albumin proteins that easily masks protein detection when running a Western blot. Therefore, we felt that a better way to address the reviewer's point is run N-terminomics instead of a Western blot using our antibodies. Also, tryptase activity cannot be seen using a Western Blot as it needs to be in a tetrameric form, and this usually dissociates when running an SDS-PAGE gel.

In the TAILS experiment, there is at least one other protease present in synovial fluid that is capable of cleaving after Arg residues (Figure 7f). Can you identify this protease? It is possible that this enzyme will also be able to cleave PRG4. This would be a key discovery as there may be more than one protease in the joint that is capable of decreasing lubrication via cleavage of PRG4.

To clarify this point, the reviewer is correct as there is an enrichment after an arginine and tryptophan residues. As there are 473 human proteases, it is highly challenging to identify or predict what proteases can cleave substrates and be active at a given time. However, we agree with the reviewer as this is an important point. Looking at our data, we identified multiple proteases capable of cleaving after an arginine (but not tryptophan) residue.

In **Supplementary Table 5**, we identified a significant increase in peptides corresponding to Complement 3, Complement factor B, and Thrombin. As the peptides from proteases were elevated in the untreated sample, those indicated potential active proteases in synovial fluid but not necessarily able to cleave PRG4. Therefore, we decided to look at other proteases that could cleave PRG4. By adding several recombinant proteases implicated in OA, like the matrix metalloproteinases (MMPs), we identified MMP3 as another protease able to cleave PRG4. See figure below.

As this protease has also been associated to be elevated in OA and play important roles in promoting OA (1- Koyama et al. (2021) *Clinical Rheumatology* 40, 2007-12; 2- Jan et al. (2021) *Frontiers in Physiology* 12: 663978; 3- Lohmander et al. (1993) *Arthritis & Rheumatology* 36(2): 181-9), we feel that it could be an important protease in the regulation of PRG4.

As we published previously (Figure 3C from Eckhard et al. (2016) *Matrix Biology* 49: 37-60), MMP3 prefers to cleave after a glutamic acid, asparagine, aspartic acid, and alanine (see image below, Figure 3C from the paper). Therefore, it is unlikely the activity observed in our N-terminomics/TAILS data. We believe that a more thorough characterization of MMP3, its activity and its substrates profile in OA should be done in a subsequent publication.

MMP3 cleavage sites in a trypsin-generated peptide library (n=208)

Another protease has been shown to be able to cleave PRG4. We have published that Cathepsin G has been shown to degrade PRG4, but the exact cleavage site remains to be characterized (Huang et al. (2020) *Scientific Reports* **10**: 4215). We identified Cathepsin G in our shotgun proteomics data; however, it was only present in 2 out of 7 samples (**Supplementary Table 3**) and no intensity were detected in the OA patient samples (**Supplementary Table 7**), therefore, we feel it is challenging to conclude that Cathepsin G is present and active in all our samples. The precise role of Cathepsin G in early OA or OA needs to be further established. To address the reviewer's point, we added two sentences in the discussion on **Page 13** mentioning Cathepsin G could be another important protease that could be implicating in regulating PRG4 in OA.

In Figure 2b, addition of Tryptase or Tryptase+ AEBSF cause and increase in the Coefficient of Friction. However, there are no controls showing that the addition of any protein would yield the same change in lubrication as is seen from PRG4+Tryptase+AEBSF. I would like to see what the addition of AEBSF alone would result in. Also, perhaps adding BSA (at the same conc as Tryptase) would be useful control.

Agreed and we did these experiments as suggested. It was an oversight on our part. We have repeated the experiment with all additional controls as shown in **New Supplementary Figure 2**. We did not see any lubricating effect when adding BSA or AEBSF alone in comparison the other controls PBS and HEPES buffer.

In Figure 2c, could earlier timepoints (e.g. 10 min and 30 min) be included to show that the coefficient of friction change is also time-dependent before 1 h.

Agreed and we did these experiments as suggested as shown in **New Figure 2c** where we added 10 min and 30 min timepoints.

Minor points:

1. In the introduction, there is a word missing in the sentence "a central mucin-like domain flanked by globular"

Corrected as suggested in the introduction on **Page 2**.

2. In the introduction, the following sentence does not make sense. "There is growing evidence that MCs play a key role in knee OA and by depleting mast cells". Do mast cells deplete mast cells??

We corrected this confusing sentence, and it is now clear what we intended to write as seen on **Page 3**.

Reviewer #2:

Unraveling molecular mechanisms of Osteoarthritis (OA) initiation and progression has been a long sought after task. Human OA is a common but yet an extremely heterogeneous disease involving infiltration of diverse inflammatory cells, loss of boundary lubrication and degradation of articular cartilage. Integrating, biochemical and omics analyses, Das et al identified mast cell tryptase β as a modulator of joint lubrication in OA which is mediated by the cleavage of

Proteoglycan 4 (PRG4) known to an important protein components for cartilage integrity. The authors monitored the degradation of PRG4 by tryptase β in vitro and characterized this process by MS using MDD model of OA in rats, and human synovial samples. OA severity could be associated with degraded PRG4 which activates NF- κ B expression in cells overexpressing TLRs. The results indicate that maintaining intact PRG4 levels are key for cartilage integrity and lubrication. Overall, the reported experiments described by the authors are logic, clear, and well interpreted. The importance of PRG4 and its degraded fragments as an inflammatory signaling molecules was recently identified. Nevertheless, the identification of mast cell derived tryptase β as key enzyme in this process is an important new discovery. Thus inhibition of tryptase β enzymatic activity and maintaining normal physiological levels of PRG4, even locally, could be a promising therapeutic strategy for OA as suggested by the authors.

Nevertheless, this manuscript will benefit from the following suggested experiments and revisions:

- 1. In vivo inhibition of tryptase β , possibly by using the anti-tryptase inhibitory antibody, may be performed in DMM model to further evaluate it as key main factor in OA progression. Alternatively, treatment of DMM model with pan tryptase β inhibitor + PRG4 should be performed to validate the therapeutic potential of this treatment suggested by the authors.*

This is a great suggestion, and we are heading in this direction; however, this goes beyond the current manuscript as there are no current selective drugs against tryptase β . There are no mouse or rat tryptase antibody-based drug that exist either. Additionally, we do not have access to the human anti-tryptase β developed by Genentech™, as it is under intellectual property, and their monoclonal antibody only reacts with human tryptase but not mouse or rat. Therefore, we would not be able to use them even if we had access to them.

Importantly, small molecule inhibitors of tryptase, including APC 366, have been terminated because of their lack of specificity, potency, or toxicity issues. (1- Rymut et al. (2022) *Clinical and Translation Science* **15**(2): 451-463; and 2- Khrisna et al. (2001) *J Allergy Clin. Immunol.* **107**: 1039-1045). Therefore, new methods of blocking tryptase in preclinical and clinical studies are urgently needed to evaluate the role of tryptase in OA or joint inflammation. In relation to this point, there are two highly relevant and interesting papers that have demonstrated that serine proteases are promoting inflammation and cartilage destruction in mouse models of OA. One paper by Wang and colleagues (Wang et al (2019) *eLife* **8**:e39905) have demonstrated increased mast cell tryptase activity in OA patients as compared to non-OA patients. The authors used the destabilization of the medial meniscus (DMM) in mice and demonstrated that mast cell deficiency in two mouse models (*Kit*^{W-sh/W-sh} and Hello *Kitty*) resulted in protection against OA-related mouse pathologies. In this publication, they also treated the mice with the non-selective tryptase inhibitor APC 366 demonstrating a decrease in inflammation, cartilage degradation, osteophyte formation and synovitis. As APC 366 is not a selective tryptase β inhibitor, we have recently started to develop a selective rat tryptase β monoclonal antibody to perform this experiment, but it will take more than 2-3 years to get this antibody therefore, it will be included in a subsequent publication.

Additionally, we have also shown previously that addition of recombinant PRG4 in the DMM model in rat resulted to increase NF- κ B (Das et al. (2019) *Bioessays* **41**(1): 1800166). See below

Figure 1: PRG4 localization in articular cartilage is negatively associated with NF- κ B staining in cartilage post-DMM. In normal rat cartilage (left panels), PRG4 staining is observed in the superficial layer, with minimal NF- κ B staining in the adjacent chondrocytes. Post-DMM, PRG4 (green) surface staining is disrupted, and this is associated with increased NF- κ B (red) staining in the adjacent chondrocytes (middle panels). If recombinant PRG4 is introduced into the post-DMM rat joint, surface staining of PRG4 is restored with a concurrent decrease in NF- κ B staining (right panels). DAPI (blue) nuclear stain, scale bars equal 50 μ m.

2. The authors should specify the expression system of the recombinant proteins and their glycosylation profiles compared to the physiological proteins.

Agreed. Structural and functional characterization of the PRG4 used in this study was done by author Tannin Schmidt where he demonstrated that it has similar higher order structure and O-linked glycosylation to that of endogenous PRG4 as shown in Samson M et al. (2014) *Experimental Eye Research* **127**: 14-19. Full-length human recombinant PRG4 (rhPRG4), produced by Lubris BioPharma, used in our study was generated using Chinese hamster ovary (CHO) cells. This information has now been added in the material and methods section with this reference on **Page 14**.

We have also demonstrated that the PRG4 used here can adsorb to articular cartilage and function as an efficient lubricant further indicating its ability to retain native glycosylation as shown in Abubacker S et al. (2016) *Connective Tissue Research* **57**(2): 113-123.

Different glycosylation patterns may explain the differences in the cleavage sites identified in recombinant vs ex vivo PRG4.

This is a great point raised by the reviewer. Yes, we think it is likely an explanation of why the cleavage sites would differ between a healthy and OA synovial fluid. This information has been added to the discussion on **Pages 12-13**:

“We also found a cleavage site at ¹³⁰⁶K↓A¹³⁰⁷ (**Fig. 1f**) using ATOMS that was identified in OA patients’ synovial fluid using N-terminomics (**Fig. 8**). Importantly, PRG4’s glycosylation is not identical between healthy and OA patients (Estrella, et al. (2010) *Biochem. J.* **429**, 359–367); and may differ to CHO cells O-glycosylated recombinant form (Huang et al. (2020) *Sci. Rep.* **10**, 4215). Therefore, a simultaneous kinetic characterization of the glycosylation profiles and proteolysis of PRG4 in synovial fluids would be of interest to better understand its function in OA.”

To further expand on this point, we demonstrated in **Supplementary Figure 1a**, that deglycosylation of PRG4 resulted in more efficient degradation of PRG4 suggesting that the mucin domain and glycosylation of PRG4 protects or partially prevent cleavage of PRG4. Importantly, in **Figure 1** and **Supplementary Figure 1**, we added two recombinant proteins in one tube, therefore, it is reasonable to expect faster cleavage and degradation of a substrate. If you look at *in vivo* models or *ex vivo* treatment of PRG4, we expect to have a different physiological range of protein levels including protease inhibitors and proteases at physiological conditions. Importantly, the cleavage sites of PRG4 identified in **Figure 7** and **new Figure 8** were also identified using our mass spectrometry approach ATOMS shown in **Figure 1e-g** and **Supplementary Figure 1b-c**.

Are the fragments remained glycosylated. Is glycosylation responsible for NF-κB? The authors should elaborate on this point in text.

This is an interesting point and we have elaborated on this in the discussion. As shown in **Figure 7j**, our data demonstrated that adding trypsin alone to human synovial fluid is sufficient to induce a change in gene expression in *NF-κB1* and *RELA*. Therefore, we believe that glycosylation is not a requirement for NF-κB activation but is likely implicated. Also, we previously demonstrated that altered truncated PRG4 glycans can stimulate the synoviocyte secretion of vascular endothelial growth factor a (VEGFA), IL8 and CCL3 (Huang et al. (2022) *Frontiers in Molecular Biosciences* **9**: 942406). However, it is likely that different glycosylation profiles of PRG4 between healthy and OA patients’ synovial could contribute to changes in NF-κB activation. For example, there is a report (Ramakrishnan et al. (2013) *Science Signaling* **6**(290): 1-13) that demonstrated that the addition of O-linked β-N-acetylglucosamine, a process also called O-GlcNAcylation, can modify the NF-κB subunit c-REL resulting in O-GlcNAcylation activation. Serine350 was reported to be the site of O-GlcNAcylation. Therefore, this point raised by the reviewer warrants further investigation and it should be tested whether trypsin alone is sufficient to induce NF-κB activity or O-GlcNAcylation or other forms of glycosylation can also regulate NF-κB activity.

3. Was tryptase β deglycosylated prior to treatment of synovial fluids? In Materials and methods, Fig 1b, d. Fig. 2b: need to specify the initial concentrations of PRG4.

Tryptase was not deglycosylated as it is not a glycoprotein. There is one amino acid that could be glycosylated at amino acid 233 but no biological implications have been made (information taken from <https://www.uniprot.org>). The point of the deglycosylation was to study PRG4 as it is heavily glycosylated especially in the mucin-like domain. Therefore, we thought it was not necessary to deglycosylated tryptase before the addition to synovial fluids.

We decided to report a protease to protein molar ratio of 1:10, 1:100 and 1:1000 as it is an important information. We apologize for the oversight, and we have now added the initial mass of PRG4 used for these experiments and it was 1 μ g. This information has been added to the **material and methods** and **Figures 1b, d** and **Figure 2b** and **c**.

4. Include image analysis in Fig.3 to show decrease/increase of PRG4 for both domains. In addition, statistical significance based on chosen analysis should be presented.

Agreed and all the data analysis has been performed as seen in **New Supplementary Figures 5** and **6**. The experimental details were also added in the material and methods.

Reviewer #3:

This study by Das and colleagues examined the cleavage of proteoglycan-4 (PRG4) by beta-tryptase as it pertains to osteoarthritis. Specific cleavage sites and resulting products of proteolysis were identified. The effect of tryptase-induced PRG4 cleavage resulted in a reduction in the lubricating and joint protective properties of the glycoprotein. Combined inhibition of tryptase activity with PRG4 rescued the beneficial effects of PRG4. This is an interesting study which has been well executed and highlights a potential new treatment strategy for osteoarthritis.

1. Inhibition of tryptase activity with AEBSF improves the lubrication coefficient of PRG4 (Fig. 2B); however, it does not recover to PRG4 alone levels. The authors should discuss what other factors could be cleaving PRG4 to limit its lubricating properties.

We have performed additional experiments and added more controls to **Supplementary Figure 2**. We believe that AEBSF or BSA are not impacting the lubricating properties of PRG4. Therefore, we think that it could be that AEBSF does not block 100% of the activity of tryptase β . Another explanation is that the cleaved PRG4 fragments could also potentially impact PRG4 lubricating properties as PRG4 is known to form multimers and the cleaved fragments could be interfering with this.

2. In Fig. 3: does the “uninjured control” refer to sham-operated animals? If so please change. If not please add in a sham group. It would be nice to have a semi-quantification of the expression levels between the different groups (e.g. area of immunoreactivity).

Yes, they refer to sham-operated animals. We have changed and updated all the figure panels and legends. We have updated the text and replaced “uninjured” to sham-operated control.

3. The authors should demonstrate that inhibition of tryptase activity improves the protective effect of PRG4 on joint damage (i.e. OARSI scores).

This is a great suggestion, and we are heading in this direction; however, this goes beyond the current manuscript as there are no selective drugs against tryptase β . There are no mouse or rat tryptase antibody that exist. Additionally, we do not have access to the human anti-tryptase β developed by Genentech, as it is under intellectual property, and their monoclonal antibody only reacts with human tryptase and not mouse or rat. Therefore, we would not be able to use them if we had access to them. Small molecule inhibitors of tryptase, including APC 366, have been terminated because of their lack of specificity, potency, or toxicity issues. (1- Rymut et al. (2022) *Clinical and Translation Science* **15**(2): 451-463; and 2- Khrisna et al. (2001) *J Allergy Clin. Immunol.* **107**: 1039-1045). Therefore, new methods of blocking tryptase in preclinical and clinical studies are urgently needed to evaluate the role of tryptase in OA or joint inflammation.

In relation to this point, there are two highly relevant and interesting papers that have demonstrated that serine proteases are promoting inflammation and cartilage destruction in mouse models of OA. One paper by Wang and colleagues (Wang et al (2019) *eLife* **8**:e39905) have demonstrated increased mast cell tryptase activity in OA patients as compared to non-OA patients. The authors used the destabilization of the medial meniscus (DMM) in mice and demonstrated that mast cell deficiency in two mouse models (*Kit*^{W^{-sh}/W^{-sh} and Hello *Kitty*) resulted in protection against OA-related mouse pathologies. In this publication, they also treated the mice with the non-selective tryptase inhibitor APC 366 demonstrating a decrease in inflammation, cartilage degradation, osteophyte formation and synovitis. As APC 366 is not a selective tryptase β inhibitor, we have recently started to develop a selective rat tryptase β monoclonal antibody to perform this experiment, but it will take more than 1-2 years to get this antibody therefore, it will be included in a subsequent publication.}

We have also shown previously that addition of recombinant PRG4 in the DMM model in rat resulted to increase NF- κ B (Das et al. (2019) *Bioessays* **41**(1): 1800166). See below

Figure 1: PRG4 localization in articular cartilage is negatively associated with NF-κB staining in cartilage post-DMM. In normal rat cartilage (left panels), PRG4 staining is observed in the superficial layer, with minimal NF-κB staining in the adjacent chondrocytes. Post-DMM, PRG4 (green) surface staining is disrupted, and this is associated with increased NF-κB (red) staining in the adjacent chondrocytes (middle panels). If recombinant PRG4 is introduced into the post-DMM rat joint, surface staining of PRG4 is restored with a concurrent decrease in NF-κB staining (right panels). DAPI (blue) nuclear stain, scale bars equal 50 μm.

We have also previously demonstrated that PRG4 (or lubricin) injection improved OARSI scores and maintained cartilage health (Iqbal et al. (2016) *Scientific Reports* 6: 18910).

Figure S8. Lubricin injection maintains cartilage health. One week after DMM surgery either saline (A) or 200 μg/kg lubricin (B) was injected into the joints of the rats. The rats injected with lubricin demonstrated significantly better cartilage histological (OARSI grading) outcomes, lower osteophyte scoring, and lower synovial inflammation grading (C). Lubricin expression was detected at the surface of the cartilage and meniscus in saline injected and lubricin injected animals. + = Synovitis, ^ = Osteophyte, # = Cartilage degeneration, p=0.05. Scale bar equals 200 μm.

4. *The study would benefit greatly from some sort of functional read-out (e.g. improved gait, reduced inflammation in vivo).*

Great suggestion. As mentioned in the previous point, we have previously demonstrated that PRG4 (or lubricin) injection improved OARSI scores and maintained cartilage health (Iqbal et al. (2016) *Scientific Reports* **6**: 18910). For the current study, we ran a new experiment to demonstrate the benefit of PRG4 injection as shown in **NEW Figure 3i**. We ran a Luminex cytokine/chemokine array to compare saline and PRG4 injections in rat subjected to a DMM model. We found that injections of PRG4 decreased the levels of TNF α and IL6 but not G-CSF, GM-CSF, IL1 β , IFN γ and CCL2. Therefore, by tracking decreased levels of TNF α and IL6, we generated a new functional readout *in vivo*.

5. *When examining the expression of PRG4 in synovial fibroblasts, why did the experimenters change to the GPI model of inflammatory joint disease? Shouldn't this be carried out in the DMM model for consistency?*

Yes, we agree with the reviewer. We have added two additional datasets. In **new Supplementary Figure 9**, we mined human OA fibroblasts (GSE176308). In **new Supplementary Figure 10**, we mined murine knee synovium following a joint trauma and contributes to post-traumatic osteoarthritis (PTOA) progression (GSE211584). However, when examining all currently available datasets, no existing datasets contains expression of *Prg4*, *Tlr2*, *Tlr4*, *Tlr5* and *Thy1* within the same dataset. Therefore, we decided to keep our GPI model and complemented our previous analysis with two additional datasets further supporting a suggestive interacting role between PRG4 and TL4 but not TLR2 and TLR5. Additionally, we found another dataset (Zhao et al. (2022) *Clinical Immunology* **244**: 109117) that would have been ideal as it was done in the destabilized medial meniscus (DMM) mouse model of OA but the RAW data were not submitted therefore, they cannot be mined or analyzed. Therefore, within the reach of our capability, we added two additional datasets analyses still supporting a interacting role between TLR4 and PRG4.

6. *The sex, strain, and weight range of animals used in the study must be described.*

Agreed and it was added in the material and methods. We have used a total of 8 female and 5 male 10-week-old Lewis rats. In **Figure 3**, we are showing female rat pictures and in the **Supplementary Figures 3-4**, we are showing male rat pictures. The males weighed ~250g +/- 50g and females weighed 175g +/- 20g at the time of surgery.

We have also added the OA patients and healthy subjects in **NEW Supplementary Figures 12-13** and in the material and methods. An ethic statement was also added.

7. *Add page numbers for ease of review.*

Agreed and it has now been added.

REVIEWERS' COMMENTS

Reviewer #1 (Remarks to the Author):

The authors have addressed all of my concerns.

Reviewer #2 (Remarks to the Author):

The revised manuscript by Das et al is an improved version over the original manuscript. The authors invested time and efforts in revising the manuscript according to the referees suggestions and comments when possible. In this referee view it would be great if in-vivo studies could be generated but as no specific inhibitors against tryptase β are available to the authors, this task seems to be beyond the scope of their work. Nevertheless, I trust that the message of this manuscript and the great amount of proteomic data generated on human samples will be of significance to the field. Overall, this work supports the conclusion and the presented results are of high quality. PRG4 is a key protein components for cartilage integrity and therefore its processing by tryptase β is of great importance for drug discovery.

I recommend accepting this manuscript for publication.

Reviewer #3 (Remarks to the Author):

The additional cytokine data are welcome but not what I had in mind. A physiological readout would have been more precise. Nevertheless, these additional cytokine data support the general point.

Our answer to the reviewers:

Reviewer #1 (Remarks to the Author):

The authors have addressed all of my concerns.

Thank you.

Reviewer #2 (Remarks to the Author):

The revised manuscript by Das et al is an improved version over the original manuscript. The authors invested time and efforts in revising the manuscript according to the referees suggestions and comments when possible. In this referee view it would be great if in-vivo studies could be generated but as no specific inhibitors against tryptase β are available to the authors, this task seems to be beyond the scope of their work. Nevertheless, I trust that the message of this manuscript and the great amount of proteomic data generated on human samples will be of significance to the field. Overall, this work supports the conclusion and the presented results are of high quality. PRG4 is a key protein components for cartilage integrity

*and therefore its processing by tryptase β is of great importance for drug discovery.
I recommend accepting this manuscript for publication.*

Thank you for the feedback and positive comments.

Reviewer #3 (Remarks to the Author):

The additional cytokine data are welcome but not what I had in mind. A physiological readout would have been more precise. Nevertheless, these additional cytokine data support the general point.

Thank you and we aim to continue validating these findings in other animal models and human biopsies with extensive physiological readouts, cytokine analysis, proteomics analysis and antibody validations.

Sincerely,

Antoine Dufour

Emails: antoine.dufour@ucalgary.ca

Phone: 1-403-210-7426

Cumming School of Medicine

Physiology and Pharmacology

University of Calgary

Calgary, AB, Canada